# WINIQ: ACCELERATING QUANTIZATION-AWARE TRAINING OF LLMS AROUND SADDLE POINTS

## ABSTRACT

Quantization-aware training is a widely used approach for language model quantization in sub-4-bit precision. This approach works by training full-precision weights to minimize the loss with gradients on the quantized model. Despite its superior performance, the main bottleneck for this quantized training is its slow convergence, which gets worse in lower bit-widths. While this problem has been observed in prior work, its precise cause has not been carefully studied. In this paper, we analyze the convergence by computing the Hessian spectrum of the model loss throughout quantization-aware training. We find the key reason is that the model weights converge to flat surfaces near saddle points with a large fraction of Hessian eigenvalues concentrated around zero, and the magnitude of both positive and negative eigenvalues decreases over training. Additionally, the convergence speed is slower in lower bit-widths with significantly smaller magnitude of loss Hessian eigenvalues. Motivated by these findings, we propose an approach to accelerate quantized training with minimal overhead named WINIQ. The key technique in WINIQ is periodical weight re-initialization by linear interpolation between the full-precision and quantized weights. This interpolation resets the weights to regions with larger (magnitude) Hessian eigenvalues without increasing the loss. We further use noise injection to regularize the Hessian, resulting in an algorithm that is broadly applicable to quantization methods. Extensive experiments show that WINIQ accelerates various quantized training methods by up to $4\times$. Under the same training budget as prior training methods, WINIQ improves state-of-the-art sub-4-bit quantization performance by up to **8.8**% relatively. Additionally, WINIQ remains consistently effective across 16 settings of different language models, quantization methods, and bit-widths.

## 1 INTRODUCTION

Quantization is a key technique for efficiently deploying large language models by representing weights and activations with lower precision. Since directly converting a trained model to lower precision often increases loss, quantization-aware training (QAT) (Jacob et al., 2018) was introduced to reduce the loss caused by quantization. This approach works by training the full-precision model weights while simultaneously applying quantization to the weights and, in some cases, to activations. This training approach has significantly advanced the language model performance in extremely low precision. Recent developments of quantized training have led to language models in sub-4-bit precision with close performance to full-precision (Liu et al., 2025b; Panferov et al., 2025), while post-training quantization methods remain significantly less accurate below 4-bit precision (Ashkboos et al., 2024; Liu et al., 2025a).

A major bottleneck of quantization-aware training is slow convergence and early stagnation in high-loss regions, which leads to substantial computational cost and limits further improvement. For example, starting from a pretrained model, the training cost of quantized training under 4-bit remains around 10% of full-precision pretraining (Liu et al., 2025b). Training with lower bit-width quantization, such as 1-bit, is even more difficult, as the convergence becomes remarkably slower (Wang et al., 2023). It normally requires training on at least 10B tokens for 1-bit precision, even on language models with 100M parameters (Panferov et al., 2025). Further increasing the training cost brings limited gains by current scaling laws (Kumar et al., 2025). This work aims to understand the reason behind the slow convergence and propose methods to accelerate quantized training for

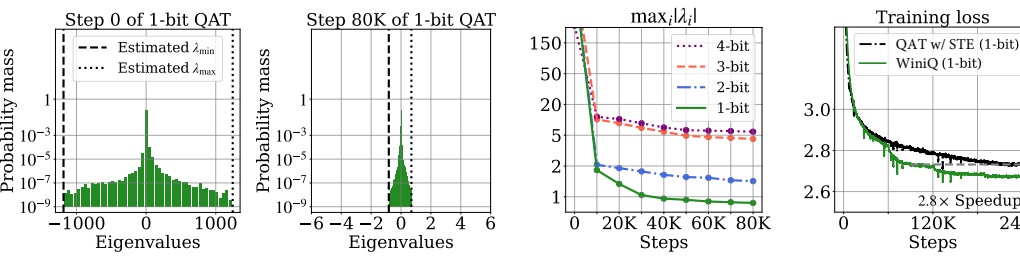

(a) The estimated empirical eigenvalue distribution of the loss Hessian at step 0 and 80K in 1-bit QAT

(b) Maximum magnitude of Hessian eigenvalues

(c) Comparing our method with the state-of-the-art

Figure 1: We investigate the slow convergence of quantization-aware training by analyzing the loss Hessian with respect to model weights. Figure 1a: By estimating the empirical distribution of the Hessian eigenvalues, we find that slow convergence is mainly because the model weights converge to a flat surface around saddle points, where the rate is governed by the magnitude of Hessian eigenvalues. In low precision, over 40% of the Hessian eigenvalues are near zero, and the maximum absolute eigenvalue decreases markedly during training. Figure 1b: We further observe that at lower bit-widths, Hessian eigenvalues have much smaller magnitudes, consistent with the slower convergence in these settings. Figure 1c: We propose WINIQ, which integrates a novel weight re-initialization technique with noise injection. Our approach applies to various quantized training methods with minimal overhead and significantly speeds up SoTA methods in extremely low-precision.

language models. Our objective is to reduce the cost required to reach a given training loss, and to achieve better loss under the same budget.

Most prior works have focused on improving the design of quantization methods and gradient estimation in quantized training, while the problem of slow convergence has not been studied in depth. Uniform quantization is the most widely used quantization method. Due to its non-differentiability, a technique named straight-through estimation (STE) (Bengio et al., 2013) is often used to estimate gradients by copying the gradients of quantized weights to the full-precision weights. Based on this prior practice, a recent work, ParetoQ (Liu et al., 2025b), reduces the error in weight quantization with a new stretched elastic quantization method and learnable step sizes (Esser et al., 2020). QuEST (Panferov et al., 2025) improves the quantization error with a normalizing Hadamard Transform and refines gradient estimation by masking the gradients of weights with high quantization errors. Yet, improving the slow convergence of training has been left open in prior works.

In this work, we examine the slow convergence of quantization-aware training by analyzing the Hessian of the quantized model loss. Our motivating observation is that the gradient norm of quantized training often drops near zero, suggesting that the model weights are converging to a first-order stationary point where the training speed is governed by second-order gradients. Thus, we hypothesize that in low precision, the loss Hessian matrix exhibits many small–or even zero–eigenvalues, thereby slowing down the convergence.

To this end, we first estimate the empirical eigenvalue distribution of the loss Hessian for quantized language models, using numerical methods based on Hessian–vector products (Bai et al., 1996; Ghorbani et al., 2019). Our main finding is that across 1- to 4-bit training, model weights converge to a flat surface around saddle points. As illustrated in Figure 1a, the Hessian exhibits a balanced mix of positive and negative eigenvalues, with over 40% concentrated near zero. Provided that the gradient norm is also approaching zero, this indicates that the weights are stuck around saddle points, where the convergence speed is governed by the maximum magnitude of Hessian eigenvalues. After the initial training phase, the magnitude of eigenvalues decreases remarkably, which causes slow convergence. Further, as shown in Figure 1b, the magnitude of eigenvalues becomes even smaller at lower bit-widths, consistent with the slower convergence observed in lower-bit precision. Our findings establish a novel connection between the slow convergence of quantized training and the saddle point problem in high-dimensional non-convex optimization (Dauphin et al., 2014).

A common approach to accelerate training is using second-order optimization methods, which often involve extensive computational cost for estimating the Hessian. In this work, we design a simple and fast approach for QAT with negligible additional cost. Our approach is based on a novel re-

initialization technique where model weights are periodically reset to an interpolation between $W$ and $Q(W)$, denoted as $(1 - \alpha)W + \alpha Q(W)$ for an $\alpha$ between 0 and 1. This design is motivated by the observation that the maximum magnitude of Hessian eigenvalues tends to be larger along the interpolation, thus speeding up the convergence. Further, we incorporate a noise injection method that computes the gradient after perturbing $W$ with a random Gaussian noise $U$ at each iteration, as inspired by prior theoretical work (Jin et al., 2017) for accelerating training around saddle points. Combining the two techniques, our approach, named WINIQ, can be applied on top of various quantization methods, including those using the Hadamard Transform.

We extensively evaluate WINIQ across multiple bit-widths and language models. First, at precisions below 4-bit, our method achieves **1.5–4**× speed-ups over state-of-the-art quantized training methods, with one example shown in Figure 1c. Second, under the same computational cost, our approach advances the sub-4-bit quantization performance by as much as **8.8**%. Across 16 different settings, we observe consistent gains for LLaMA and Qwen models ranging from 0.6B to 3B parameters. Third, WINIQ can also be applied to quantization methods with the Hadamard Transform and further boosts the performance by up to **2.8**%. Finally, ablation studies confirm that both components in our algorithm are essential to the performance. The code for reproducing this work can be found at https://anonymous.4open.science/r/WiniQ.

In summary, this paper makes three contributions to quantized training for low-precision LLMs:

- First, we approach the slow convergence of quantized training through analyzing the loss Hessian and identify that the primary reason arises from model weights converging to a flat surface near saddle points.
- Second, we propose a generic approach to accelerate quantized training with negligible computational overhead, combining a novel weight re-initialization technique and noise injection.
- Third, we demonstrate the broad applicability of our approach across quantization methods. It both significantly accelerates the convergence and advances the state-of-the-art performance in sub-4-bit quantization for language models.

## 2 PRELIMINARIES

In this paper, we study quantization-aware training for sub-4-bit quantization of language models. We describe a general formulation. Let $f_W$ denote a language model with parameters $W \in \mathbb{R}^d$. Let $\hat{L}_W$ denote its empirical risk on a training set. Let $Q(\cdot)$ be a quantization function that maps the model parameters to weights that can be represented by lower precisions. We denote its output weights as $Q(W)$. Quantization-aware training minimizes the loss $\hat{L}_{Q(W)}$ with respect to $W$ through computing $Q(W)$ at each iteration. Thus, $W$ is also called latent weights.

As $Q(\cdot)$ is typically non-differentiable, QAT algorithms often use the straight-through estimator, which copies the gradient of $Q(W)$ as the gradient of $W$. After training, the final latent weights $\hat{W}$ are quantized to $Q(\hat{W})$ for inference. We provide precise definitions of other terminologies, such as saddle points, in Appendix A. Next, we describe the widely used uniform quantization, which can be instantiated as many quantization methods.

**Definition 2.1** (Uniform quantization). Given a bit-width $n$, uniform quantization linearly scales the weights and rounds them to the nearest integers:

$$Q(W) = a \left\lfloor \text{clip}\left(\frac{W - b}{a}, v_{\text{neg}}, v_{\text{pos}}\right) \right\rceil + b, \tag{1}$$

where $a$ is the scale, $b$ is the bias, and $\lfloor \rceil$ is a rounding function that converts the value to its nearest integer. $\text{clip}(W, v_{\text{neg}}, v_{\text{pos}})$ clamps the values in $W$ to a range between $v_{\text{neg}}$ and $v_{\text{pos}}$, where the bounds are determined by the quantization bits.

For more concrete examples, in symmetric min-max quantization, $a$ is set by the maximum absolute weight: $a = \frac{\max_i |W_i|}{2^{n-1}-1}$, with $v_{\text{neg}} = -2^{n-1}$, $v_{\text{pos}} = 2^{n-1} - 1$, and $b = 0$. In asymmetric min-max quantization, the scale is based on the full range of weights: $a = \frac{\max_i W_i - \min_i W_i}{2^n - 1}$, with $v_{\text{neg}} = 0$, $v_{\text{pos}} = 2^n - 1$, and $b = \min_i W_i$. In addition, the scale $a$ can also be treated as a learnable parameter and optimized jointly with the model (Esser et al., 2020).

**Convergence rate under varied bit-width.** We present empirical observations of the convergence speed using a state-of-the-art QAT method based on STE (Liu et al., 2025b). We perform quantization-aware training for the pretrained LLaMA-3-1B on the FineWebEdu dataset, using AdamW as the optimizer.

In Figure 2, we first illustrate the convergence of QAT in 1-bit, 2-bit, 3-bit, and 4-bit, respectively. We find that the convergence slows down after the initial 10K training steps. QAT in 2-bit and 1-bit converges significantly slower than 4-bit quantiza-

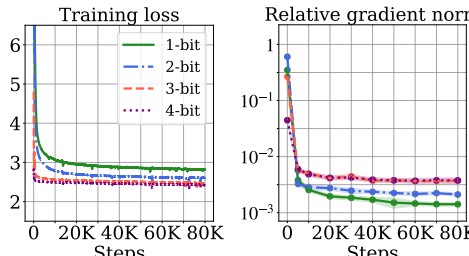

Figure 2: The convergence of the training loss and the relative gradient norm in quantization-aware training from 1-bit to 4-bit precision.

tion. Further, we evaluate the gradient norm relative to the weight norm, i.e., $\|\nabla_W \hat{L}_{Q(W)}\|_2 / \|W\|_2$, along training, which converges to the scale between $10^{-3}$ and $10^{-2}$ across 1-bit to 4-bit precision.

Near a first-order stationary point, the convergence rate of gradient-based methods is governed by the magnitude of Hessian eigenvalues. One hypothesis is that low-bit quantization tends to yield flat curvature in the weight space. Additionally, we observe that the distance between $W$ and $Q(W)$ increases in lower precision, suggesting the increasing difficulty of training in lower precision. The relative norm of quantization error, i.e., $\|Q(W) - W\|_2 / \|W\|_2$, increases from 16% in 4-bit to 70% in 1-bit. This motivates us to study the connection between the slow convergence of quantized training and flat curvature in the loss surface.

## 3 OUR APPROACH

In this section, we analyze the Hessian spectrum of the loss surface during quantized training. We find that the Hessian exhibits concentrated zero eigenvalues with small magnitudes of both negative and positive eigenvalues. Thus, the slow convergence can be attributed to the existence of saddle points, which are more severe in lower precision. To tackle this, we discover that a weight re-initialization technique significantly speeds up training by linearly interpolating weights with quantized weights. We propose an approach that further incorporates noise injection into model weights during training, which generally applies to and speeds up various quantization methods.

### 3.1 MEASURING HESSIAN SPECTRUM

We first present our analysis of the loss Hessian with regard to model weights. Specifically, we estimate the empirical spectral density of the Hessian matrix, i.e., the eigenvalue and its probability mass over all eigenvalues. Fully computing the Hessian matrix for a large model is infeasible; we use a numerical tool named Stochastic Lanczos Quadrature (Bai et al., 1996; Chen et al., 2021), which can be efficiently implemented via repeatedly computing Hessian-vector products. We present results with LLaMA-3-1B during quantized training with 1-bit to 4-bit precision and estimate the loss Hessian on the training data. See further implementation details in Appendix A.

**Saddle point problem in quantization-aware training.** We identify that the model weights get stuck in a flat region near saddle points. Figure 3 illustrates the estimated eigenvalue distributions in 1-bit, 2-bit, and 3-bit weight precision, respectively. Consistently across three precisions, we observe that there are both negative and positive eigenvalues in the Hessian matrix with comparable probability mass. As training progresses, more eigenvalues concentrate near zero, and the magnitude of eigenvalues decreases significantly. For example, in 3-bit QAT, the probability mass of zero eigenvalues increases from **7**% at step 0 to **41**% at 80K steps[1]. See Appendix A for a precise definition of saddle points.

**Flatter curvature in lower bit-width.** We further find that the model weights are stuck in a region with flatter curvature for lower precision, resulting in slower convergence of quantization-aware training. In Figure 3, comparing among the three bit-widths, we found that the magnitude of the eigenvalues becomes much smaller, from **6** in 3-bit to **2** in 2-bit and less than **1** in 1-bit at 80K

---

[1]Numerically, we regard the estimated eigenvalues in a range between $-10^{-3}$ and $10^{-3}$ as zero eigenvalue.

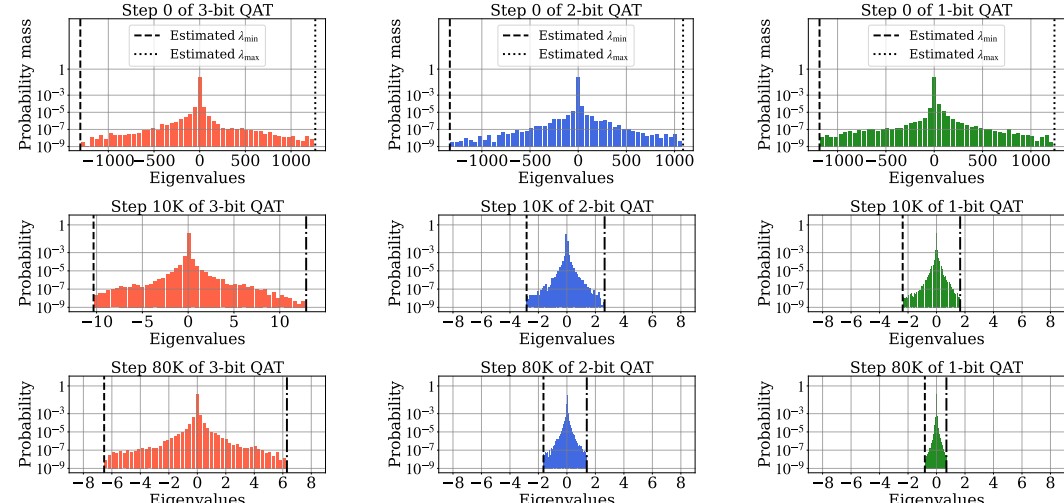

Figure 3: We illustrate the estimated eigenvalues and their probability mass of the loss Hessian matrix in 3-bit, 2-bit, and 1-bit QAT, respectively. (1) Most of the eigenvalues concentrate close to zero, and there are both negative and positive eigenvalues with comparable probability mass. This suggests that model weights converge to a flat surface around the saddle points during training. (2) For a lower precision, the magnitude of eigenvalues becomes smaller, and more eigenvalues are around zero. This suggests that the model weights are in a region with flatter curvature, thus exhibiting slower convergence. The $x$-**axis** corresponds to the eigenvalues. $y$-**axis** corresponds to the probability mass function of the eigenvalues (in log scale). We illustrate the eigenvalues for the weights in transformer layers. We describe the results for the full precision model and for embedding layers in Appendix B.

steps. Additionally, a larger proportion of eigenvalues is near zero. The probability mass for zero eigenvalues increases from **41**% in 3-bit to **55**% in 2-bit and **63**% in 1-bit.

Additionally, we further evaluate the Hessian spectrum for 8-bit and 4-bit activation quantization using Llama-1B with 1-bit weight quantization. We find that training with lower-precision activations (8-bit and 4-bit) produces (**23**% and **32**%) more smaller-magnitude Hessian eigenvalues of model weights compared to 16-bit activations, respectively. This indicates a flatter loss landscape and correlates with the slower training under low-precision activation quantization.

Additionally, we find that the loss Hessian in the embedding layer exhibits a different eigenvalue distribution than transformer layers, with most eigenvalues being non-negative. In contrast, for transformer layers in full precision, we still observe a balanced mix of positive and negative eigenvalues. In experiments, we only quantize the weights in transformer layers, while keeping the embedding layer in full precision, following protocols in prior works. Since the majority of model parameters lie in the transformer layers, the saddle point problem mainly hinders convergence. Additional evaluations of the analysis are provided in Appendix B.

## 3.2 ALGORITHM DESIGN

Our findings suggest that the key to improving the convergence of quantized training is to tackle the saddle point problem. Next, we present a fast algorithm that significantly accelerates training in low precision, with minimal computational overhead.

**(1) Weight re-initialization.** Our key technique is to reset the latent weights $W$ to the linear interpolation between $W$ and $Q(W)$ during training:

$$W \leftarrow (1 - \alpha)W + \alpha Q(W), \text{ given a scalar } \alpha \in [0, 1]. \tag{2}$$

This is motivated by the finding that the Hessian eigenvalue magnitudes (measured by the maximum absolute value) increase for interpolated weights. As illustrated in Figure 4, we estimate the

**Algorithm 1** WINIQ: **W**eight (re)-**i**nitialization with **n**oise **i**njection for **Q**uantization-aware training

---

**Input**: Initialization $W_0 \in \mathbb{R}^d$, a quantization function $Q(\cdot)$, a language model $f_W$
**Require**: A re-initialization interval $K$, an interpolation scalar $\alpha$, standard deviation $\sigma$, number of iterations $T$ and learning rates $\eta$
**Output:** The trained latent weights $W_T$

1: **for** $i = 0, 1, \ldots, T-1$ **do**
2: $\quad U_i \leftarrow$ Sample a random noise from $\mathcal{N}(0, \sigma^2 \operatorname{Id}_d)$
3: $\quad W_{i+1} \leftarrow W_i - \eta \nabla_{Q(W_i + U_i)} \hat{L}_{Q(W_i + U_i)}$
4: $\quad$ **if** $i + 1 (\operatorname{mod} K)$ is zero **then**
5: $\quad\quad W_{i+1} \leftarrow (1 - \alpha)W_{i+1} + \alpha Q(W_{i+1})$
6: $\quad$ **end if**
7: **end for**

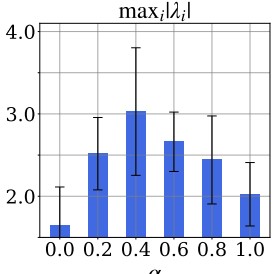

Figure 4: We illustrate the increase in the maximum magnitude of Hessian eigenvalues at the interpolated weights relative to $\alpha$.

Hessian eigenvalues on the interpolated weights with $\alpha$ from 0 to 1, using the LLaMA-1B trained at 2-bit precision. We find that $\alpha = 0.4$ yields **84**% larger magnitude of eigenvalues than that at $W$. Additionally, the probability mass of the zero eigenvalues is decreased by **21**%. We present a visualization of the effect in Figure 10. We have consistent observation of the increase of Hessian eigenvalue magnitude across various quantization settings from 1-bit to 4-bit. The results are described in Appendix B. Correspondingly, our experiments find that, across language models and bit widths, the interpolation accelerates the convergence of the training loss.

Thus, we propose to re-initialize the latent weights periodically during training. Specifically, given a scalar $\alpha \in [0, 1]$ and an interval of $K$ steps, we re-initialize the latent weights $W$ to the linear interpolation weights every $K$ steps by Equation 2. We note that the re-initialization step does not change the second-order states in the optimizer. Thus, loss curvature information accumulated by the AdamW optimizer is preserved.

**(2) Noise injection.** Second, we incorporate a noise injection method in the training of latent weights. At each iteration, we inject a random Gaussian noise $U \sim \mathcal{N}(0, \sigma^2 \operatorname{Id})$ into the latent weights $W$ and compute gradients on $Q(W + U)$. This is inspired by prior works that show that performing SGD with noise-perturbed weights improves the convergence of gradient descent for non-convex optimization near saddle points (Jin et al., 2017). In experiments, we find that noise injection also speeds up quantization-aware training. Taken together, we describe our approach, named WINIQ, in Algorithm 1.

**Extension to incorporate Hadamard Transform.** Our approach can be applied to quantization methods that use the Hadamard Transform (Panferov et al., 2025). These methods multiply the weight vector at each layer with a Hadamard matrix before applying a quantization function, which can reduce the quantization error. Given a matrix $H \in \mathbb{R}^{d \times d}$ as a block-diagonal matrix whose diagonal blocks are per-layer Hadamard matrices, we can extend the re-initialization step as:

$$W \leftarrow H^\top \left((1 - \alpha)HW + \alpha Q(HW)\right), \tag{3}$$

based on the fact that $H$ is an orthogonal matrix. This can be efficiently computed with fast Hadamard multiplication kernels (Dao et al., 2025). The extended approach with the Hadamard Transform is described in Appendix A.

**Discussion about computational overhead.** Our method can be applied on top of existing quantization-aware training algorithms with negligible overhead. The re-initialization step computes the quantized weights and interpolates them with the latent weights. This involves simple operations like addition and multiplication, incurring less than 1% of the computational cost of the language model forward pass. Moreover, this step is performed only every $K$ iterations, where $K$ is typically large (e.g., one-fourth of total training steps), making the cost negligible compared to training. For the noise injection, our approach samples one noise vector every step, using the same amount of forward and backward passes as in the base training method.

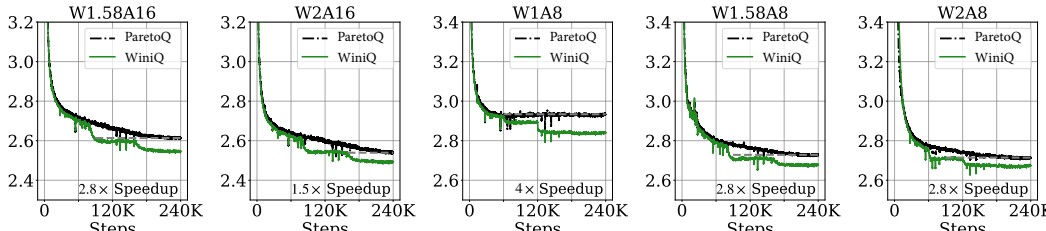

Figure 5: We evaluate training loss convergence by comparing our approach with the SoTA STE-based quantization-aware training method on the LLaMA-3-1B model. Our method accelerates convergence by **1.5-4×** across weight precisions of 1–2 bits and activation precisions of 8–16 bits. In higher-precision settings, longer re-initialization intervals generally lead to improved performance. Throughout, we use the notation W2A16 to indicate 2-bit weights and 16-bit activations, with analogous notation for other configurations. The result of W1A16 is reported in Figure 1.

## 4 EXPERIMENTS

In this section, we evaluate WINIQ across diverse weight and activation precisions for multiple language models. First, we show that our approach speeds up the convergence of SoTA quantized training methods by up to **4×**, including the challenging setting of 1-bit weights and 8-bit activations. Second, at the same training cost, WINIQ delivers up to **8.8**% improvement over leading baselines, evaluated across ten bit-width configurations and four language models. Further, we show that our approach applies to another quantization method with the Hadamard Transform, achieving up to **2.3**% gains over the SoTA performance in the 1-bit weight and 4-bit activation setting. Finally, ablation studies confirm the contribution of both components of our approach.

### 4.1 EXPERIMENTAL SETUP

Our experiments evaluate quantization-aware training with sub-4-bit weight quantization, including 4, 3, 2, 1.58, and 1 bit, where 1.58-bit corresponds to ternary quantization. We further combine these weight quantization settings with activation precisions of 16, 8, and 4 bits.

**Datasets and models.** We perform QAT on various language models, including LLaMA-3-1B, LLaMA-3-3B, Qwen-3-1.7B, and Qwen-3-0.6B. We perform quantized training on a language modeling dataset, FineWebEdu. We train each model up to 20B tokens, typically in 240K steps. After training, we evaluate quantized language models on several downstream tasks, following prior works. These include a language modeling dataset, WikiText2, and eight QA datasets, including ARC-easy, ARC-challenge, BoolQ, PIQA, SIQA, HellaSwag, OBQA, and WinoGrande. The sources for the models and datasets are described in Appendix B.

**Baselines.** We compare our algorithm with existing quantization methods for language models. We mainly compare our algorithm against two state-of-the-art QAT methods, including ParetoQ (Liu et al., 2025b) and QuEST (Panferov et al., 2025). Besides, we compare with post-training quantization (PTQ) methods to illustrate the relative improvement of QAT methods. These include Round-to-Nearest (RTN), GPTQ (Frantar et al., 2023), AWQ (Lin et al., 2024), and SpinQuant (Liu et al., 2025a). We also include the performance of the full-precision pretrained model for reference.

**Implementations.** We implement our approach on top of existing QAT methods, adopting the same quantization functions. Specifically, we use Elastic Binarization (Liu et al., 2022) for 1-bit weights, Stretched Elastic Quant (Liu et al., 2025b) for 1.58 and 2-bit weights, and LSQ (Esser et al., 2020) for 3 and 4-bit weights. Activations are quantized using symmetric quantization. For comparisons with QuEST, we follow its setup, incorporating the Hadamard Transform, MSE-optimal fitting, and the trust gradient estimator. As in prior methods, we quantize the non-embedding weights.

Our approach includes three key hyperparameters: the re-initialization interval $K$, the interpolation scalar $\alpha$, and the noise standard deviation $\sigma$. We vary $K$ across 40K, 60K, and 80K for a total of 240K steps; $\alpha$ between 0.1 and 0.6; and $\sigma$ between 0.0002 and 0.002. The training uses the AdamW optimizer with a learning rate in the range of $1 \times 10^{-5}$ to $4 \times 10^{-5}$. We discuss the hyperparameter tuning in Section 4.3 and report the hyperparameters used for each result in Appendix B.

Table 1: This table reports the comparison of our approach to recent quantization methods. We evaluate the quantization performance across various settings on LLaMA models, with weights in 1 to 4 bits and activations in 8 to 16 bits. We report the perplexity (PPL) on WikiText2 and the average zero-shot test accuracy (Acc.) across eight QA datasets. To show relative performance, we also evaluate the full-precision (FP) pretrained model. W2A16 means 2-bit weights and 16-bit activations, and analogous notations for others. 1.58-bit means ternary quantization.

| | LLaMA-1B, W1A16 | | LLaMA-1B, W1.58A16 | | LLaMA-1B, W2A16 | | LLaMA-1B, W3A16 | | LLaMA-1B, W4A16 | |
|---|---|---|---|---|---|---|---|---|---|---|
| Metrics | PPL ($\downarrow$) | Acc. ($\uparrow$) | PPL ($\downarrow$) | Acc. ($\uparrow$) | PPL ($\downarrow$) | Acc. ($\uparrow$) | PPL ($\downarrow$) | Acc. ($\uparrow$) | PPL ($\downarrow$) | Acc. ($\uparrow$) |
| FP Model | 9.6 | 58.5 | 9.6 | 58.5 | 9.6 | 58.5 | 9.6 | 58.5 | 9.6 | 58.5 |
| RTN (PTQ) | 4.2e8 | 33.7 | 1.8e6 | 36.2 | 1.5e6 | 38.5 | 30.9 | 38.8 | 13.9 | 52.4 |
| GPTQ (PTQ) | 3.3e8 | 32.7 | 4.6e4 | 32.8 | 3.3e2 | 36.8 | 68.6 | 41.1 | 13.4 | 52.8 |
| AWQ (PTQ) | - | - | - | - | 2.0e5 | 36.4 | 1.5e2 | 42.0 | 12.2 | 56.4 |
| SpinQuant (PTQ) | 2.4e8 | 33.7 | 2.2e3 | 32.6 | 46.7 | 38.3 | 12.6 | 51.9 | 10.3 | 56.5 |
| ParetoQ | 16.9 | 51.9 | 14.0 | 54.7 | 12.5 | **56.7** | 10.9 | 57.2 | 10.3 | **58.7** |
| WINIQ | **15.3** | **52.6** | **12.9** | **55.6** | **11.9** | 56.6 | **10.9** | **57.8** | **10.2** | 58.6 |
| | LLaMA-1B, W1A8 | | LLaMA-1B, W1.58A8 | | LLaMA-1B, W2A8 | | LLaMA-3B, W1A8 | | LLaMA-3B, W1.58A8 | |
| FP Model | 9.6 | 58.5 | 9.6 | 58.5 | 9.6 | 58.5 | 7.7 | 65.2 | 7.7 | 65.2 |
| RTN (PTQ) | 4.7e8 | 33.7 | 1.8e6 | 36.2 | 1.5e6 | 36.1 | 7.3e7 | 33.7 | 7.9e5 | 33.2 |
| GPTQ (PTQ) | 3.8e8 | 31.7 | 7.5e4 | 32.7 | 3.8e4 | 32.7 | 5.9e7 | 32.8 | 2.7e5 | 33.1 |
| SpinQuant (PTQ) | 3.4e8 | 32.8 | 5.8e3 | 32.7 | 3.8e2 | 34.9 | 4.5e7 | 32.8 | 3.1e3 | 33.3 |
| ParetoQ | 23.3 | 48.2 | 18.2 | 51.9 | 16.9 | 52.2 | 15.7 | 54.0 | 13.1 | 55.9 |
| WINIQ | **21.9** | **49.0** | **16.9** | **52.5** | **16.3** | **53.0** | **14.8** | **55.2** | **12.2** | **58.6** |

## 4.2 EXPERIMENTAL RESULTS

**Comparing training costs.** First, we compare the training cost of our approach with the state-of-the-art method, ParetoQ, for weights in 1–2 bits and activations in 8–16 bits. The speed-up rate is measured by the reduction in training steps to reach the same error. As shown in Figure 5, WINIQ consistently outperforms ParetoQ across six bit-width settings, with larger acceleration at lower precisions. For weight-only quantization, WINIQ achieves **1.5–2.8**× speed-up from 2-bit to 1-bit. With both weights and activations quantized, it delivers up to a **4**× speed-up in the challenging setting of 1-bit weights and 8-bit activations. Furthermore, WINIQ converges to a lower training loss, improving upon the asymptotic performance of the current state-of-the-art.

**Comparing quantization performance.** We present the quantization performance of low-precision weights (1–2 bits) and activations (8–16 bits) on LLaMA models in Table 1. Our approach advances the state-of-the-art across eight weight and activation quantization settings. For weight-only quantization, WINIQ surpasses the strongest baseline by up to **8.8**% in PPL and **1.6**% in average zero-shot accuracy, while also markedly outperforming all existing PTQ methods. When both weights and activations are quantized, it achieves relative gains of **7.1**% in PPL and **1.6**% in accuracy over the best baseline. On larger models such as LLaMA-3B, WINIQ delivers a **6.8**% relative improvement. Consistent with the convergence results, these gains are most pronounced at lower precisions.

We observe that our approach yields larger performance gains at precisions below 3 bits. At higher bit-widths like 4-bit, existing quantized model performance is close to the full-precision model, leaving limited room for improvement. We include these results for completeness, and our approach does not degrade performance at higher precisions.

**Extensions.** We extend our approach to the Qwen models and observe consistent improvements over the state-of-the-art baseline. As shown in Table 2, WINIQ achieves up to a **3.2**% relative reduction in PPL and a **1.3**% gain in zero-shot test accuracy compared to ParetoQ.

We further demonstrate the broad applicability of our approach to different quantization methods. When combined with another state-of-the-art method (Panferov et al., 2025) using Hadamard Transform (Dao et al., 2025), WINIQ improves performance by **2.8**% in 1-bit weight and 4-bit activation quantization. The results are shown in Table 3.

## 4.3 ABLATION STUDIES

We discuss the main hyperparameters of WINIQ, including the interpolation scalar $\alpha$, a re-initialization interval $K$, and the standard deviation $\sigma$ in noise injection.

Table 2: We report the quantization performance of applying our approach to Qwen models, as compared to the SoTA method, ParetoQ. We evaluated the models in sub-2-bit weights and 8-bit activations.

|  | Qwen-1.7B, W1A8 | | Qwen-0.6B, W1A8 | |
|---|---|---|---|---|
|  | PPL ($\downarrow$) | Acc. ($\uparrow$) | PPL ($\downarrow$) | Acc. ($\uparrow$) |
| FP Model | 16.2 | 58.1 | 16.2 | 58.1 |
| ParetoQ | 46.5 | 42.2 | 64.0 | 41.2 |
| WINIQ | **45.6** | 42.2 | **61.9** | **41.4** |
|  | Qwen-1.7B, W2A8 | | Qwen-0.6B, W2A8 | |
| ParetoQ | 22.2 | 47.8 | 32.0 | 43.3 |
| WINIQ | **21.8** | **48.2** | 32.0 | **43.9** |

Table 3: We report the quantization performance of our approach with Hadamard Transform on LLaMA-1B, applied on top of another SoTA method, QuEST. We evaluate the model in sub-2-bit weights and 4-bit activations.

|  | LLaMA-3-1B, W1A4 | |
|---|---|---|
|  | PPL ($\downarrow$) | Acc. ($\uparrow$) |
| FP Model | 9.6 | 58.5 |
| QuEST | 42.9 | 42.4 |
| WINIQ w/ Hadamard Transform | **42.3** | **43.0** |
|  | LLaMA-3-1B, W2A4 | |
| QuEST | 17.4 | 48.6 |
| WINIQ w/ Hadamard Transform | **16.9** | **49.3** |

**Effect of interpolation on training speed.** We first examine the effect of the interpolation scalar $\alpha$ in weight re-initialization. Recall that values of $\alpha$ around 0.4 tend to yield the largest increase in the maximum absolute eigenvalues of the Hessian. We vary $\alpha$ when training the LLaMA-1B model in 1-bit precision and observe the training loss curves. As shown in Figure 6, we observe that $\alpha$ around 0.4 leads to the fastest convergence after re-initialization, corresponding to the increase in Hessian eigenvalue magnitude.

**Influence of each hyperparameter of our approach.** Further, we perform an ablation study that varies each hyperparameter of our approach for training the LLaMA-1B model in 1-bit precision. The results are shown in Table 4. Compared to training without noise ($\sigma = 0$), noise injection improves performance by about 4%. Compared to training without weight re-initialization ($\alpha = 0$), weight re-initialization improves performance by about 6.7%. This suggests that both component of our approach contributes to the final performance. Moreover, we find that smaller values of $\alpha$ and larger re-initialization intervals tend to yield the best results.

**Strategies for selecting hyperparameters.** Lastly, we discuss the strategies for choosing the hyperparameters. Recall that interpolation is defined as $(1 - \alpha)W + \alpha Q(W)$, which reduces $\|W - Q(W)\|_2$ by a factor of $\alpha$. After two re-initializations, this distance is approximately $(1-\alpha)^2$ of its original value. Smaller values of $\alpha$ keep the weights closer to the latent weights. Empirically, we find that re-initializing too close to either the latent or quantized weights results in suboptimal performance. To balance this trade-off, we choose relatively small $\alpha$ values from 0.1 to 0.4 and perform at most three re-initializations, typically every 60K or 80K steps out of 240K steps.

For the standard deviation $\sigma$ in noise injection, we find that the optimal $\sigma$ can vary across models. In practice, we find that $\sigma = 0.001$ typically works the best for LLaMA models, and $\sigma = 0.0002$ works the best for Qwen models. We report the hyperparameters used in every experiment in Table 16 of Appendix B.

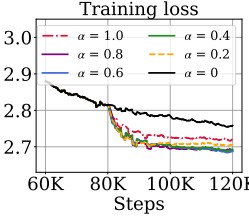

| PPL ($\downarrow$) | Weight re-initialization | | | Noise injection |  |
|---|---|---|---|---|---|
|  | $K = 40K$ | $K = 60K$ | $K = 80K$ | $\alpha = 0.4, K = 60K$ |  |
| $\alpha = 0.0$ | 16.5 | 16.4 | 16.7 | $\sigma = 0$ | 16.0 |
| $\alpha = 0.2$ | 15.6 | 15.5 | 15.5 | $\sigma = 0.0005$ | 15.7 |
| $\alpha = 0.4$ | 16.1 | **15.3** | 15.6 | $\sigma = 0.001$ | **15.3** |
| $\alpha = 0.6$ | 16.2 | 15.5 | 15.8 | $\sigma = 0.002$ | 16.3 |
| $\alpha = 0.8$ | 16.3 | 16.0 | 16.1 | $\sigma = 0.004$ | 18.5 |

Figure 6: We show the convergence of training loss after re-initialization with various $\alpha$. This is evaluated in training LLaMA-3-1B with 1-bit weight quantization. $\alpha = 0$ indicates no re-initialization.

Table 4: We report the performance of varying hyperparameters in our approach, including the interpolation scalar $\alpha$, re-initialization interval $K$, and standard deviation $\sigma$. We train LLaMA-3-1B with 1-bit weight quantization and report the perplexity (PPL) on the WikiText-2 dataset. $\sigma = 0$ indicates no noise injection. $\alpha = 0$ means no weight interpolation.

## 5 RELATED WORK

Second-order derivatives have provided key information for designing quantization methods. For example, the error in quantization can be approximated using a second-order Taylor expansion. Based on this, OBQ (Frantar & Alistarh, 2022) and GPTQ (Frantar et al., 2023) design efficient algorithms to iteratively perform layerwise quantization using an estimated Hessian matrix. QuIP (Chee et al., 2023) proposes incoherence processing that minimizes a proxy objective derived from the second-order approximation through adaptive rounding. In these works, the loss Hessian is typically estimated from the second-order moments of input features. By computing Hessian top eigenvalues with Hessian-vector products (Yao et al., 2020), HAWQ (Dong et al., 2019) proposes a mixed-precision quantization method that uses the top eigenvalues as the layerwise sensitivity measure. Further, using the Hessian trace as a sensitivity measure has improved mixed-precision performance (Dong et al., 2020). In contrast, our work examines the eigenvalue distribution of the Hessian during quantized training, revealing behaviors different from other deep learning settings (Yao et al., 2018). Our analysis suggests that the key challenge for quantization-aware training is addressing the saddle point problem. Further, the Hessian statistics have been associated with the generalization of deep neural networks (Ju et al., 2022; Zhang et al., 2024a). It raises an interesting future direction to analyze the generalization of quantized models based on the loss Hessian.

Quantized training has been widely used for low-bit quantization (Yin et al., 2024). In the context of language models, LLM-QAT (Liu et al., 2024) proposes a distillation method that performs QAT on data generated by pretrained models. One focus of prior works is on improving the gradient estimation in low-precision settings. Quant-Noise (Fan et al., 2021) proposes to only quantize a random subset of weights at each training iteration. More recently, Fifty et al. (2025) proposes a rotation trick that rotates and linearly rescales the gradients to improve gradient estimation for vector quantization. QuEST (Panferov et al., 2025) proposes a new gradient estimator by masking the gradients on weights with large quantization errors, which improves the scaling laws of language models in sub-4-bit quantization (Kumar et al., 2025). Our contribution is to connect the slow convergence of QAT to saddle point problems and design a general approach that accelerates QAT methods with negligible overhead. We refer the readers to a survey for a more comprehensive review on quantization-aware training in other areas like computer vision (Gholami et al., 2022).

An active line of research has focused on designing faster optimizers to accelerate the training of large language models. One approach leverages layerwise adaptive learning rates with large batch sizes (You et al., 2020). Another direction is to improve the pre-conditioner design in Adam. For example, the Lion optimizer (Chen et al., 2023), discovered through program search over lightweight gradient-based pre-conditioners, accelerates Adam in training transformers. Liu et al. (2023) propose a lightweight second-order optimizer that uses stochastic estimates of the Hessian diagonal as a pre-conditioner, which speeds up pretraining GPT models. In contrast, our work focuses on accelerating quantization-aware training.

## 6 CONCLUSION

This work studies the slow convergence in quantization-aware training for language models. Our analysis of the loss Hessian revealed that model weights tend to be stuck in a flat region near saddle points, especially at lower precision. This insight led us to develop a simple method to accelerate quantization-aware training based on a weight re-initialization technique combined with noise injection. Our approach demonstrates significant speed-up and improvement of quantization performance over various methods. These findings provide an improved understanding of quantization-aware training from an optimization perspective and open up many new directions for designing scalable training methods for LLMs in low-precision.

Developing theoretical guarantees for quantized training is challenging because assumptions in the existing theory may not hold under quantization. For instance, straight-through estimators can produce abrupt gradient changes that violate smoothness assumptions. Developing formal convergence guarantees is an interesting direction for future research, and our work provides an empirical foundation to help guide such efforts.

## REPRODUCIBILITY STATEMENT

To facilitate the reproduction of the experimental results presented in this paper, we provide a detailed description of the algorithmic procedure in Section 3. The datasets, models, and hyperparameter tuning strategies in our experiments are discussed in Section 4. All datasets and models used in this work are publicly accessible online. Their respective sources are documented in Table 10 of Appendix B, and the exact hyperparameters corresponding to each reported result are provided in Table 16 of Appendix B. Furthermore, we are committed to open-sourcing the implementation to enable full reproducibility after the review process.

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

## A DETAILS OF OUR APPROACH

### A.1 NOTATIONS AND TERMINOLOGIES

To precisely describe the concepts used in this paper, we introduce the notations and definitions following the terminology from Jin et al. (2017). For a function $f : \mathbb{R}^d \to \mathbb{R}$, we use $\nabla f(\cdot)$ and $\nabla^2 f(\cdot)$ to denote its gradient and Hessian. For vectors we use $\|\cdot\|$ to denote the $\ell_2$-norm. We use $\lambda_{\max}(\cdot), \lambda_{\min}(\cdot), \lambda_i(\cdot)$ denote the maximum, minimum, and $i$-th eigenvalues.

We begin by formalizing the smoothness property of functions, which ensures that the gradient does not change too rapidly.

**Definition A.1** ($\ell$-smooth)**.** A differentiable function $f : \mathbb{R}^d \to \mathbb{R}$ is said to be $\ell$-smooth (or $\ell$-gradient Lipschitz) if

$$\|\nabla f(x_1) - \nabla f(x_2)\| \leq \ell \|x_1 - x_2\|, \quad \forall x_1, x_2 \in \mathbb{R}^d.$$

Then, we define stationary points, which are critical in analyzing the convergence of gradient descent methods.

**Definition A.2** (First-order stationary point)**.** For a differentiable function $f(\cdot)$, we say that $x$ is a *first-order stationary point* if $\|\nabla f(x)\| = 0$. We also say $x$ is an $\varepsilon$-*first-order stationary point* if

$$\|\nabla f(x)\| \leq \varepsilon.$$

A first-order stationary point can be either a local minimum, a saddle point, or a local maximum. We describe the definition of local minima as follows.

**Definition A.3** (Local minimum). For a differentiable function $f(\cdot)$, a point $x$ is a *local minimum* if it is a first-order stationary point and there exists $\delta > 0$ such that

$$f(x) \leq f(y), \quad \forall y \in B(x, \delta),$$

where $B(x, \delta)$ denotes the ball of radius $\delta$ centered at $x$.

Finally, not all stationary points are desirable in optimization. In particular, a point may be stationary but not a local minimum. We define saddle points as follows.

**Definition A.4** (Saddle point). For a differentiable function $f(\cdot)$, a point $x$ is a *saddle point* if it is a first-order stationary point but not a local minimum. For a twice-differentiable function $f(\cdot)$, a saddle point $x$ is *strict* (or non-degenerate) if $\lambda_{\min}(\nabla^2 f(x)) < 0$.

In our evaluations, we observe that the model weights in quantization-aware training converge to an approximate first-order stationary point, while the corresponding loss Hessian exhibits more than one negative eigenvalue. This indicates that the model weights are around saddle points.

A.2 COMPUTING THE HESSIAN SPECTRUM

In this section, we briefly describe the Stochastic Lanczos Quadrature (SLQ) algorithm (Bai et al., 1996), which we use to estimate the empirical eigenvalue distribution of the loss Hessian. The method samples the spectrum using random probe vectors and computes the spectrum information through the Lanczos algorithm into a small tridiagonal matrix, whose eigenvalues approximate those of the Hessian.

Let $H$ denote a large symmetric matrix (e.g., the Hessian of a loss function with respect to model weights). The spectral density is defined as:

$$p(\lambda) = \frac{1}{n} \sum_{i=1}^{n} \delta(\lambda - \lambda_i)$$

where $\lambda_i$ are the eigenvalues of $H$ and $\delta$ denotes the Dirac delta function. Since computing the Hessian matrix is infeasible in large language models, the method uses Hessian–vector products to estimate the Hessian spectrum.

The spectral density can be expressed in the form of the trace of certain matrix functions $\phi(t) = \frac{1}{n}\text{tr}(f_t(H))$ where $f_t(H) = \delta(tI - H)$. Then, one can estimate the trace through Hutchinson's method (Hutchinson, 1989) with random Rademacher or Gaussian vectors. SLQ method samples multiple random Rademacher vectors $v$, and then approximates $v^\top f_t(H)v$, which provides estimates of the spectral density.

To compute $v^\top f_t(H)v$ efficiently, the SLQ method applies $k$ iterations of the Lanczos algorithm with $H$ and a random probe vector $v$. This yields an orthonormal basis $Q_k$ for the Krylov subspace $\mathcal{K}_k(H, v)$ and a $k \times k$ tridiagonal matrix $T_k$ such that $Q_k^\top H Q_k = T_k$. Then, diagonalizing the matrix, $T_k = U\Lambda U^\top$, yields approximated eigenvalues $\{\theta_i\}$ together with associated weights $\{w_i = (U_{1,i})^2\}$ for estimating the eigenvalue distribution of $H$.

Taken together, we summarize the procedure of the SQL algorithm as follows. First, for $j = 1, \ldots, m$, draw a random Rademacher vector $v_j$. Second, run $k$-step Lanczos on $H, v_j$ to construct the tridiagonal matrix $T_k$. Diagonalize $T_k$ to obtain $\{\theta_i^{(j)}, w_i^{(j)}\}$. Lastly, aggregate the weights to approximate the Hessian spectrum as follows:

$$\frac{1}{m} \sum_{j=1}^{m} \sum_{i=1}^{k} w_i^{(j)} \delta(\lambda - \theta_i^{(j)})$$

One can further convolve the spectrum with a Gaussian kernel, obtaining a continuous estimate of the spectrum.

In our evaluation, we estimate the empirical eigenvalue distribution of the loss Hessian, using the loss of the model on $2 \times 10^5$ tokens in the training dataset. We apply the SLQ method with $m = 200$ random vectors and $k = 100$ steps. Then, we estimate the empirical distribution obtained from the $\{\theta_i^{(j)}, w_i^{(j)}\}$, where the weights are normalized to ensure the total probability mass equals one. We report the resulting discrete distribution without additional kernel smoothing. The Hessian-vector products are implemented with existing functions provided by PyTorch.

### A.3 EXTENSION

Next, we extend our approach to incorporate the Hadamard Transform for quantization-aware training. In this method, a Hadamard matrix is applied to weights and activations before quantization at each layer. We denote a matrix $H$ as a matrix shaped as a block-diagonal matrix, using the Hadamard matrices of each layer as the diagonal blocks. To integrate it, we modify the re-initialization step by multiplying the interpolated weights with the inverse Hadamard matrix, which can be efficiently computed using its transpose. The procedure is described in Algorithm 2.

---

**Algorithm 2** WINIQ with the Hadamard Transform

---

**Input**: Initialization $W_0 \in \mathbb{R}^d$, a quantization function $Q(\cdot)$, a language model $f_W$, a matrix $H$ of $d$ dimension for the Hadamard Transform
**Require**: A re-initialization interval $K$, an interpolation scalar $\alpha$, standard deviation $\sigma$, the number of iterations $T$ and learning rates $\eta$
**Output:** The trained latent weights $W_T$

1: **for** $i = 0, 1, \ldots, T - 1$ **do**
2:    $U_i \leftarrow$ Sample a random noise from $\mathcal{N}(0, \sigma^2 \operatorname{Id}_d)$
3:    $W_{i+1} \leftarrow W_i - \eta H^\top \nabla_{Q(H(W_i + U_i))} \hat{L}_{Q(H(W_i + U_i))}$   ▷ The HT is also applied to activations
4:    **if** $i + 1 (\operatorname{mod} K)$ is zero **then**
5:       $W_{i+1} \leftarrow H^\top ((1 - \alpha)HW_{i+1} + \alpha Q(HW_{i+1}))$
6:    **end if**
7: **end for**

---

## B ADDITIONAL EXPERIMENTAL RESULTS

### B.1 ADDITIONAL EVALUATIONS OF HESSIANS

**Additional evaluations of the Hessian spectrum.** Figure 7 shows the empirical eigenvalue distributions of the loss Hessian during 4-bit quantization-aware training. The results align with those observed for 3-bit quantization. We find that most eigenvalues concentrate near zero, and after 80K training steps, the maximum absolute eigenvalue is reduced to less than 10.

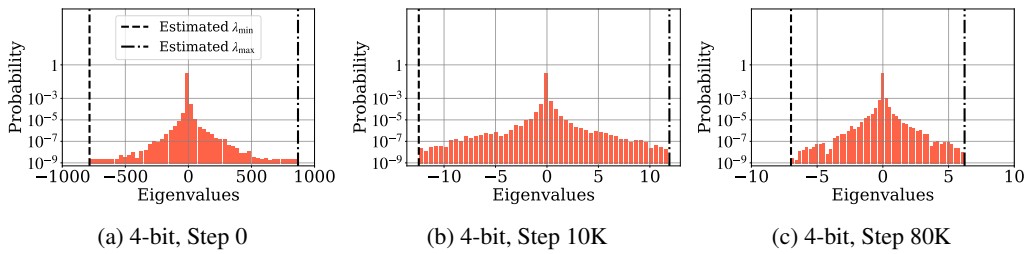

(a) 4-bit, Step 0         (b) 4-bit, Step 10K         (c) 4-bit, Step 80K

Figure 7: We plot the estimated Hessian eigenvalue distribution of the model weights at different steps of 4-bit QAT training. The $x$-axis represents the eigenvalues, and the $y$-axis shows their probability mass on a log scale.

Second, we present the empirical loss Hessian eigenvalue distributions for the embedding layer weights in Figure 8. Following common practice, we do not quantize the embedding layer. Interestingly, after training, its eigenvalues remain non-negative, in contrast to those of the transformer layers. Moreover, we observe that, at lower bit precisions, both the maximum eigenvalue and the trace (sum of eigenvalues) decrease.

Third, we illustrate the eigenvalue distribution of the weights in the transformer layers and the embedding layer, using the full-precision pretrained model. As shown in Figure 9, we observe that in the transformer layers, the Hessian shows a balanced mix of negative and positive eigenvalues, with larger magnitudes at lower precision, whereas the embedding layer consistently exhibits non-negative eigenvalues.

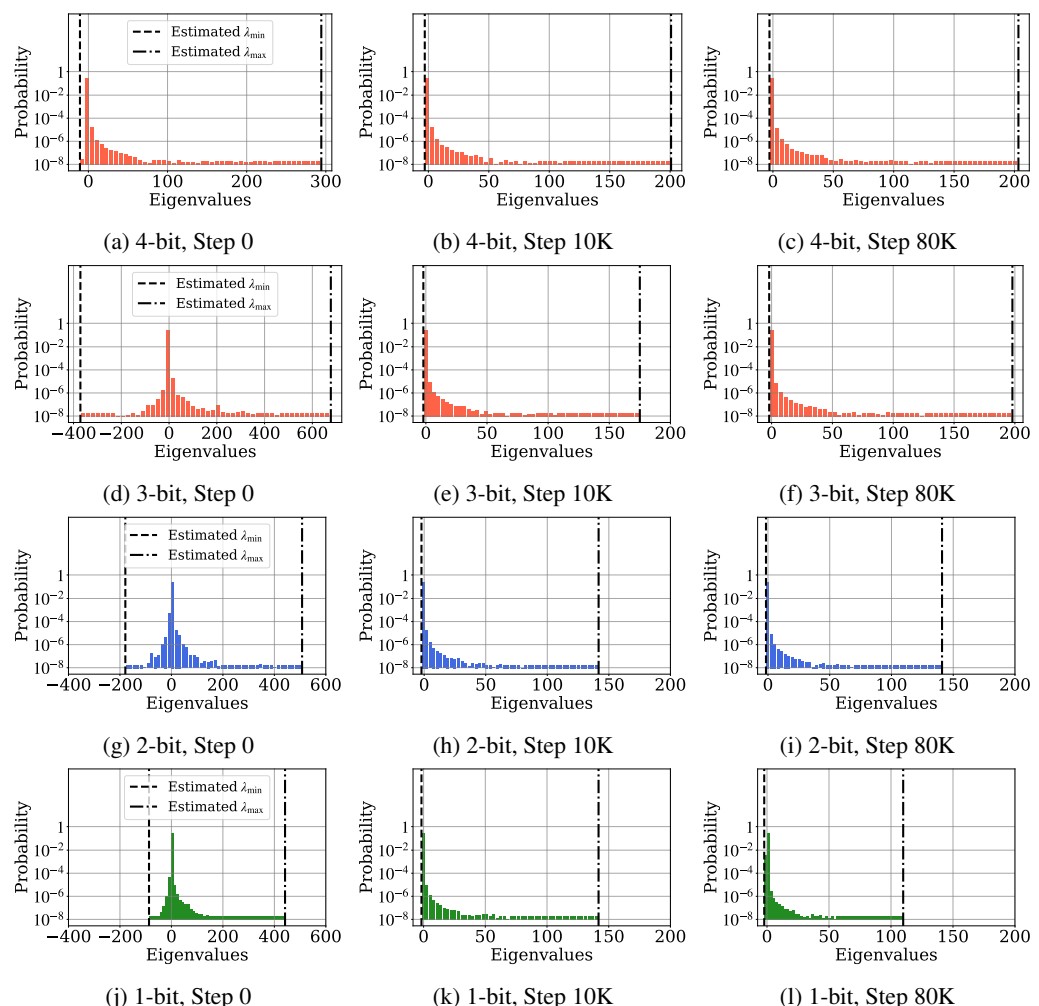

Figure 8: We plot the estimated Hessian eigenvalue distribution of the weights in the embedding layer. We illustrate the distribution for quantization-aware training with 4-bit, 3-bit, 2-bit, and 1-bit at various steps of training. The $x$-axis represents the eigenvalues, and the $y$-axis shows their probability mass on a log scale.

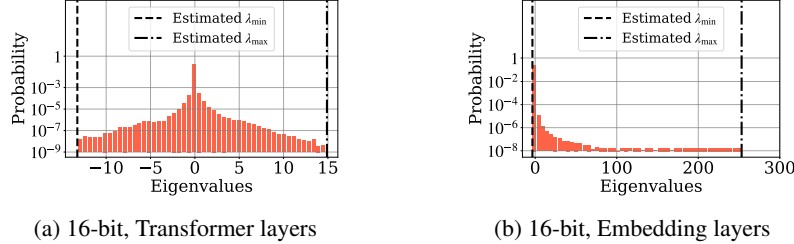

Figure 9: We plot the estimated Hessian eigenvalue distribution of the full-precision pretrained model. The eigenvalues have larger magnitudes compared to those in lower-precision settings. The $x$-axis represents the eigenvalues, and the $y$-axis shows their probability mass on a log scale.

**Evaluating Hessians when applying weight interpolation and noise injection.** Recall that shown in Figure 4, we observe that the magnitude of Hessian eigenvalues increases when evaluating interpolated weights between $W$ and $Q(W)$, which motivates our weight re-initialization technique for accelerating training.

To further support this observation, we measure Hessian eigenvalue magnitudes at different interpolation points across various quantized training settings. Using Llama-1B trained with 1–4 bit quantization at 80K steps, we compute the maximum absolute Hessian eigenvalues for interpolated weights $(1 - \alpha)W + \alpha Q(W)$ over $\alpha$ between 0.0 and 1.0. Results are reported in Table 5. We find that interpolation consistently increases the curvature, with the largest magnitudes typically occurring around $\alpha = 0.4$.

| $\max_i \|\lambda_i\|$ | $\alpha = 0.0$ | $\alpha = 0.2$ | $\alpha = 0.4$ | $\alpha = 0.6$ | $\alpha = 0.8$ | $\alpha = 1.0$ |
|---|---|---|---|---|---|---|
| 1-bit QAT | $0.93 \pm 0.3$ | $1.20 \pm 0.1$ | $1.72 \pm 0.3$ | $\mathbf{1.83} \pm 0.1$ | $1.24 \pm 0.1$ | $1.13 \pm 0.1$ |
| 2-bit QAT | $1.64 \pm 0.5$ | $2.45 \pm 0.4$ | $\mathbf{3.09} \pm 0.4$ | $2.65 \pm 0.5$ | $2.42 \pm 0.5$ | $2.05 \pm 0.5$ |
| 3-bit QAT | $6.24 \pm 0.4$ | $6.79 \pm 0.4$ | $\mathbf{7.70} \pm 0.3$ | $6.44 \pm 0.3$ | $6.68 \pm 0.3$ | $6.38 \pm 0.2$ |
| 4-bit QAT | $7.13 \pm 0.4$ | $7.31 \pm 0.4$ | $\mathbf{8.06} \pm 0.5$ | $7.61 \pm 0.5$ | $7.66 \pm 0.5$ | $7.12 \pm 0.6$ |

Table 5: We report the maximum absolute Hessian eigenvalues, $\max_i \|\lambda_i\|$, evaluated at interpolated weights $(1 - \alpha)W + \alpha Q(W)$ for Llama-1B trained with 1–4 bit weight quantization and 16-bit activations. Across all bit-widths, curvature consistently increases with interpolation and typically peaks around $\alpha = 0.4$. We compute the Hessian eigenvalues for the parameters, including the learnable quantization step sizes.

Further, we evaluate Hessian eigenvalues under noise injection. Using Llama-1B trained with 2-bit weight quantization at 80K steps, we measure the maximum absolute Hessian eigenvalues for noise standard deviations $\sigma$ between 0.0005, 0.001, and 0.002. We find that models trained with noise injection exhibit smaller negative eigenvalues (i.e., larger-magnitude curvature) than those trained without noise. In addition, noise injection produces slightly larger gradient norms. The observations explain the faster training in noise injection.

| | $\hat{L}_{Q(W)}$ | $\|\nabla_W \hat{L}_{Q(W)}\|_2 / \|W\|_2$ | $\max_i \|\lambda_i\|$ | | | | | |
|---|---|---|---|---|---|---|---|---|
| | | | $\alpha = 0.0$ | $\alpha = 0.2$ | $\alpha = 0.4$ | $\alpha = 0.6$ | $\alpha = 0.8$ | $\alpha = 1.0$ |
| $\sigma = 0$ | $3.18 \pm 0.29$ | $0.011 \pm 0.003$ | $1.64 \pm 0.5$ | $2.45 \pm 0.4$ | $\mathbf{3.09} \pm 0.4$ | $2.65 \pm 0.5$ | $2.42 \pm 0.5$ | $2.05 \pm 0.5$ |
| $\sigma = 0.0005$ | $3.08 \pm 0.28$ | $0.012 \pm 0.002$ | $3.03 \pm 0.2$ | $3.53 \pm 0.2$ | $\mathbf{3.72} \pm 0.1$ | $3.50 \pm 0.2$ | $3.40 \pm 0.2$ | $3.18 \pm 0.2$ |
| $\sigma = 0.001$ | $3.12 \pm 0.29$ | $0.013 \pm 0.002$ | $3.45 \pm 0.3$ | $3.91 \pm 0.2$ | $3.95 \pm 0.2$ | $\mathbf{3.96} \pm 0.2$ | $3.55 \pm 0.2$ | $3.32 \pm 0.7$ |
| $\sigma = 0.002$ | $3.13 \pm 0.28$ | $0.013 \pm 0.003$ | $3.53 \pm 0.2$ | $3.94 \pm 0.1$ | $3.74 \pm 0.1$ | $\mathbf{3.85} \pm 0.2$ | $3.49 \pm 0.2$ | $3.53 \pm 0.2$ |

Table 6: We report the relative gradient norms and maximum absolute Hessian eigenvalues evaluated at interpolated weights for Llama-1B trained with 2-bit quantization and 16-bit activations under varying noise standard deviation $\sigma$. Noise injection tends to yield smaller negative Hessian eigenvalues (i.e., larger absolute values) compared to standard training. We compute the Hessian eigenvalues for the parameters, including the learnable quantization step sizes.

## B.2 ADDITIONAL ABLATION STUDIES

**Jointly varying batch sizes and the noise standard deviation $\sigma$.** We conducted an ablation study jointly varying the batch size and the standard deviation $\sigma$ of the noise. We vary the batch size between 64, 32, and 16, and vary $\sigma$ between 0.0005 and 0.004. We perform the experiments using Llama-1B trained with 1-bit weight quantization and 16-bit activations for 240K steps. The results are shown in Table 7. We observe that when using smaller batch sizes (e.g., 16), smaller values of $\sigma$ tend to be better. For larger batches, using a larger $\sigma$ is better (typically around 0.001). Batch size 64 produces the best overall performance. Larger batches exceed our hardware memory limits. Accordingly, we use a batch size of 64 in the paper.

| | $\sigma = 0.0005$ | $\sigma = 0.001$ | $\sigma = 0.002$ | $\sigma = 0.004$ |
|---|---|---|---|---|
| Batch Size 64 | 15.8 | **15.3** | 16.3 | 18.5 |
| Batch Size 32 | 16.2 | 16.1 | 17.0 | 19.1 |
| Batch Size 16 | 16.6 | 16.7 | 18.2 | 20.5 |

Table 7: We report the perplexity evaluated on the WikiText2 dataset. We train Llama-1B with 1-bit weight quantization under different noise levels $\sigma$ and batch sizes. We observe that larger batch sizes benefit from using a larger $\sigma$ (around 0.001).

**Increasing the training budget.** In our main experiments, we follow the setup of prior work (Liu et al., 2025b) and train for 20B tokens (240K steps), using the same budget for our method. To further assess the scaling with respect to training budget, we doubled the budget to 40B tokens (480K steps) for Llama-1B with 1-, 2-, and 3-bit weights and 16-bit activations. As shown in Table 8, our approach continues to outperform ParetoQ at this larger budget, achieving an average perplexity improvement of **7**% on average.

| | W1A16 | | W2A16 | | W3A16 | |
|---|---|---|---|---|---|---|
| Training budget (in the number of tokens) | 20B | 40B | 20B | 40B | 20B | 40B |
| ParetoQ | 16.9 | 16.2 | 12.5 | 12.3 | 14.0 | 13.9 |
| WINIQ | **15.3** | **14.7** | **11.9** | **11.8** | **12.9** | **12.8** |

Table 8: We report the perplexity evaluated on the WikiText2 dataset, when increasing the training budget to 40B tokens. We compare WINIQ with ParetoQ under settings of W1A16, W2A16, and W3A16. WINIQ consistently outperforms the baseline after the increase of the training budget. W1A16 means 1-bit weights and 16-bit activations, and analogous notations for others.

## B.3 EXPERIMENTAL RESULTS OF 8B MODEL

Our method is not tied to any particular model size and can be applied directly to larger architectures. To explore the scalability of model sizes, we conducted an experiment on Llama-8B, training it with 2-bit weight quantization and 16-bit activations for 150K steps. As shown in Table 9, our approach achieves around a relative **1**% improvement in the average zero-shot accuracy over the state-of-the-art baseline (Liu et al., 2025b).

| | PPL ($\downarrow$) | Avg. Accuracy ($\uparrow$) | ARC-e | ARC-c | BoolQ | PIQA | SIQA | HellaSwag | OBQA | WinoGrande |
|---|---|---|---|---|---|---|---|---|---|---|
| ParetoQ | 8.4 | 64.7 | 75.9 | 50.2 | 75.0 | 78.5 | 48.5 | 72.3 | 49.6 | 68.0 |
| WINIQ | 8.3 | **65.3** | 76.2 | 52.0 | 76.9 | 78.4 | 48.5 | 72.7 | 50.4 | 67.7 |

Table 9: We report the results of training Llama-8B with 2-bit weight and 16-bit activation quantization. We report the perplexity (PPL) on WikiText2 and the average zero-shot test accuracy across eight QA datasets.

## B.4 EVALUATION OF COMPUTATIONAL OVERHEAD

The additional runtime introduced by WINIQ is negligible relative to the base training cost. We measured wall-clock time in seconds on a machine with 26 CPUs and one A100 GPU. For re-initialization, it only involves element-wise addition between two weight matrices. On Llama-1B, we perform re-initialization up to three times; This takes up to 0.3 seconds, compared to 28.7 hours for the full training run. For noise injection, each iteration samples Gaussian noise and performs an element-wise addition with the weights. On Llama-1B, this costs 0.004 seconds per iteration, which is less than 1% of the time for a forward-backward pass (0.43 seconds).

Our approach introduces no additional memory overhead. On Llama-1B, our approach uses the same peak GPU memory as the ParetoQ baseline, which is 78.8 GB. This is because memory consumption is dominated by the forward and backward passes, where our techniques do not increase the memory.

## B.5 LOSS LANDSCAPE VISUALIZATION

To further illustrate our re-initialization technique, we present a visual example of the loss surface in Figure 10. $W_t$ are located near a saddle point of the loss surface, where the curvature along two directions is close to zero, leading to stagnated training progress. In contrast, by interpolating between $W_t$ and the quantized weights $Q(W_t)$, the reinitialized weight $W_\alpha$ is reset to a region with larger Hessian eigenvalues that facilitates faster training.

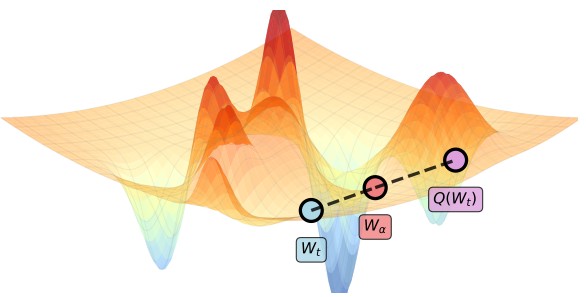

Figure 10: We illustrate an example of the loss surface with critical points that lie on a linear line. $W_t$ lies in a flat region of the surface, where the curvature is small in both directions, and training progress stalls. By interpolating between $W_t$ and its quantized counterpart $Q(W_t)$, the re-initialization moves the weight to $W_\alpha$ in a sharper region with larger Hessian eigenvalues. This transition increases Hessian eigenvalues and leads to accelerated training.

### B.6 OMITTED INFORMATION AND FULL COMPARISON RESULTS

We describe the sources for the models and datasets in Table 10. For the weight-only quantization, we describe the results in Table 11. For weight and activation quantization on Llama models, we describe the results in 12. For quantization performance on Qwen models, we describe the results in 13. For quantization performance on QuEST models, we describe the results in 14.

## C DISCUSSION ON FURTHER EXTENSIONS

In this section, we extend our study to training a model that can be quantized to multiple precisions. Specifically, given $n$ quantization functions $Q_1, Q_2, \ldots, Q_n$ (e.g., 1-bit to 4-bit quantization), our goal is to train a single model that jointly minimizes the loss across all $n$ functions. Let $L_{Q_i(W)}$ denote the expected loss of the model quantized by the $i$-th function. Our objective is then the average test loss across all quantization functions: $\frac{1}{n}\sum_{i=1}^{n} L_{Q_i(W)}$.

The motivation for this setting stems from the high computational cost of quantization-aware training, which often requires retraining separately for each quantization function. A more practical alternative is to train the model once to accommodate arbitrary bit-width quantization, enabling efficient deployment across diverse scenarios while preserving performance comparable to single-bit quantization-aware training. Such an approach would allow the creation of language models in multiple precisions, which could serve as flexible foundations for further fine-tuning on downstream tasks, for example, using high-precision low-rank adapters (Yin et al., 2024). Moreover, combining low-precision language models with KV cache quantization (e.g., Zhang et al. (2024b)) offers a promising path toward building highly efficient inference systems for state-of-the-art LLMs.

A recent work, Matryoshka Quantization (Nair et al., 2025), has explored this direction by training a single model across multiple bit-widths (2, 4, and 8 bits). Their method leverages the most significant bits of 8-bit integers to represent lower-bit integers. Conceptually, this approach can be seen as sharing clip values across different quantization levels. While their study focuses on min–max quantization, our exploration considers a more general setting that accommodates additional bit-widths and alternative methods such as LSQ.

Our preliminary study finds that noise injection stabilizes the training of multi-level bit-width QAT, yielding performance much closer to single bit-level QAT while requiring only one round of training. Specifically, we train a LLaMA-1B model jointly on 1-, 2-, 3-, and 4-bit precision under the same computational budget as training a single precision model. For each precision, we follow the quantization method in the ParetoQ method.

As shown in Table 15, the multi-bit model achieves performance comparable to single-precision models, with only a 1.7 increase in PPL and a 1.6% drop in test accuracy—while reducing training cost by 75%. For relative comparison, we evaluate against Matryoshka Quantization. Under the same training cost, noise injection delivers better results, improving average performance by reducing PPL by 8.3 and increasing test accuracy by 2.4%.

Table 10: We describe the sources for the models and datasets used in our experiments.

| Model | Source |
|---|---|
| LLaMA-3-1B | https://huggingface.co/meta-LLaMA/LLaMA-3.2-1B |
| LLaMA-3-3B | https://huggingface.co/meta-LLaMA/LLaMA-3.2-3B |
| Qwen-3-1.7B | https://huggingface.co/Qwen/Qwen3-1.7B |
| Qwen-3-0.6B | https://huggingface.co/Qwen/Qwen3-0.6B |

| Dataset | Source |
|---|---|
| FineWebEdu | https://huggingface.co/datasets/HuggingFaceFW/fineweb-edu |
| Wiki2 | https://huggingface.co/datasets/Salesforce/wikitext |
| ARC-e | https://huggingface.co/datasets/mib-bench/arc_easy |
| ARC-c | https://huggingface.co/datasets/ibragim-bad/arc_challenge |
| BoolQ | https://huggingface.co/datasets/google/boolq |
| PIQA | https://huggingface.co/datasets/ybisk/piqa |
| SIQA | https://huggingface.co/datasets/lighteval/siqa |
| HellaSwag | https://huggingface.co/datasets/Rowan/hellaswag |
| OBQA | https://huggingface.co/datasets/allenai/openbookqa |
| WinoGrande | https://huggingface.co/datasets/allenai/winogrande |

Table 11: We report the full comparison of **LLaMA-3-1B** performance under 1- to 4-bit weight quantization. We evaluate the perplexity on WikiText-2 and average zero-shot accuracy across eight QA tasks.

| | Wiki2 ($\downarrow$) | ARC-e | ARC-c | BoolQ | PIQA | SIQA | HellaSwag | OBQA | WinoGrande | Avg. Accuracy ($\uparrow$) |
|---|---|---|---|---|---|---|---|---|---|---|
| FP Model | 9.6 | 64.8 | 42.5 | 64.8 | 74.8 | 44.8 | 64.4 | 50.2 | 61.5 | 58.5 |
| **W1A16** | | | | | | | | | | |
| RTN | 4.2e8 | 25.0 | 22.5 | 37.6 | 49.5 | 32.9 | 25.0 | 27.1 | 49.6 | 33.7 |
| GPTQ | 3.3e8 | 26.9 | 21.7 | 37.6 | 51.8 | 33.5 | 25.5 | 14.8 | 49.7 | 32.7 |
| SpinQuant | 2.4e8 | 25.0 | 22.5 | 37.6 | 49.5 | 32.9 | 25.0 | 27.1 | 49.6 | 33.7 |
| ParetoQ | 16.9 | 57.3 | 36.2 | 62.4 | 69.1 | 41.9 | 48.3 | 45.1 | 55.0 | 51.9 |
| WINIQ | **15.3** | 59.7 | 37.0 | 61.3 | 69.5 | 42.6 | 49.8 | 46.1 | 54.4 | **52.6** |
| **W1.58A16** | | | | | | | | | | |
| RTN | 1.8e6 | 24.5 | 22.6 | 62.4 | 52.7 | 33.4 | 25.4 | 18.2 | 50.2 | 36.2 |
| GPTQ | 4.6e4 | 25.1 | 22.5 | 38.0 | 53.1 | 32.7 | 25.7 | 15.6 | 49.5 | 32.8 |
| SpinQuant | 2.2e3 | 25.0 | 21.5 | 37.6 | 51.9 | 33.4 | 25.3 | 17.2 | 49.1 | 32.6 |
| ParetoQ | 14.0 | 64.1 | 38.7 | 60.3 | 71.8 | 43.6 | 55.1 | 46.3 | 58.1 | 54.8 |
| WINIQ | **12.9** | 65.0 | 39.7 | 62.1 | 72.9 | 44.3 | 56.2 | 47.1 | 57.4 | **55.6** |
| **W2A16** | | | | | | | | | | |
| RTN | 1.5e6 | 26.5 | 26.8 | 62.2 | 51.0 | 36.8 | 25.9 | 28.5 | 50.2 | 38.5 |
| GPTQ | 3.3e2 | 29.3 | 27.6 | 37.8 | 51.5 | 38.6 | 26.5 | 32.0 | 50.8 | 36.8 |
| AWQ | 2.0e5 | 27.4 | 26.0 | 48.9 | 50.2 | 37.0 | 25.7 | 24.4 | 51.5 | 36.4 |
| SpinQuant | 46.7 | 25.6 | 24.6 | 62.4 | 51.6 | 36.1 | 25.8 | 29.1 | 50.8 | 38.3 |
| ParetoQ | 12.5 | 64.8 | 41.7 | 62.8 | 73.1 | 44.0 | 56.6 | 52.0 | 58.5 | **56.7** |
| WINIQ | **11.9** | 65.0 | 42.5 | 62.5 | 73.8 | 43.2 | 58.4 | 48.1 | 59.1 | 56.6 |
| **W3A16** | | | | | | | | | | |
| RTN | 30.9 | 28.9 | 25.0 | 55.9 | 53.5 | 37.8 | 30.1 | 28.9 | 50.6 | 38.8 |
| GPTQ | 68.6 | 37.4 | 27.3 | 43.1 | 58.4 | 39.2 | 37.1 | 32.4 | 53.8 | 41.1 |
| AWQ | 1.5e2 | 41.5 | 26.7 | 49.2 | 58.0 | 41.4 | 34.9 | 31.8 | 52.8 | 42.0 |
| SpinQuant | 12.6 | 56.9 | 34.9 | 61.0 | 69.3 | 42.0 | 53.4 | 41.2 | 56.2 | 51.9 |
| ParetoQ | **10.9** | 65.3 | 41.9 | 64.2 | 73.8 | 43.9 | 61.3 | 47.7 | 59.5 | 57.2 |
| WINIQ | **10.9** | 65.9 | 43.2 | 63.9 | 74.4 | 44.8 | 61.2 | 49.0 | 60.2 | **57.8** |
| **W4A16** | | | | | | | | | | |
| RTN | 13.9 | 55.7 | 36.3 | 61.9 | 70.4 | 43.0 | 56.9 | 39.3 | 55.5 | 52.4 |
| GPTQ | 13.4 | 55.2 | 38.8 | 57.9 | 70.5 | 43.5 | 55.4 | 43.2 | 58.0 | 52.8 |
| AWQ | 12.2 | 63.4 | 40.0 | 63.5 | 73.4 | 44.5 | 60.5 | 45.8 | 60.3 | 56.4 |
| SpinQuant | 10.3 | 62.2 | 40.3 | 64.1 | 72.3 | 44.0 | 61.6 | 47.9 | 59.8 | 56.5 |
| ParetoQ | 10.3 | 67.4 | 43.4 | 64.4 | 74.8 | 44.4 | 63.5 | 50.4 | 61.4 | **58.7** |
| WINIQ | **10.2** | 67.4 | 43.8 | 65.1 | 75.0 | 43.5 | 63.2 | 48.6 | 62.1 | 58.6 |

Table 12: We report the complete comparison results of **LLaMA-3-1B** and **LLaMA-3-3B** with 1–2 bit weights and 8-bit activations. We evaluate the perplexity on WikiText-2 and average zero-shot accuracy across eight QA tasks.

| LLaMA-3-1B | Wiki2 (↓) | ARC-e | ARC-c | BoolQ | PIQA | SIQA | HellaSwag | OBQA | WinoGrande | Avg. Accuracy (↑) |
|---|---|---|---|---|---|---|---|---|---|---|
| FP Model | 9.6 | 64.8 | 42.5 | 64.8 | 74.8 | 44.8 | 64.4 | 50.2 | 61.5 | 58.5 |
| **W1A8** | | | | | | | | | | |
| RTN | 4.7e8 | 25.0 | 22.5 | 37.6 | 49.5 | 32.9 | 25.0 | 27.1 | 49.6 | 33.7 |
| GPTQ | 3.8e8 | 26.9 | 21.7 | 37.6 | 51.8 | 33.5 | 25.5 | 14.8 | 49.7 | 32.7 |
| SpinQuant | 3.4e8 | 27.2 | 21.1 | 37.6 | 52.6 | 33.5 | 25.6 | 14.3 | 50.5 | 32.8 |
| ParetoQ | 23.3 | 55.4 | 31.4 | 61.2 | 66.5 | 40.5 | 38.2 | 39.8 | 52.5 | 48.2 |
| WiniQ | **21.9** | 56.0 | 33.9 | 61.6 | 66.4 | 41.6 | 38.8 | 41.4 | 52.3 | **49.0** |
| **W1.58A8** | | | | | | | | | | |
| RTN | 1.8e6 | 24.5 | 22.3 | 62.4 | 52.7 | 33.4 | 25.4 | 18.4 | 50.2 | 36.2 |
| GPTQ | 7.5e4 | 24.8 | 22.2 | 38.0 | 52.2 | 32.5 | 25.4 | 17.0 | 49.4 | 32.7 |
| SpinQuant | 5.8e3 | 25.3 | 22.5 | 37.6 | 51.6 | 33.3 | 25.3 | 17.6 | 48.5 | 32.7 |
| ParetoQ | 18.2 | 59.6 | 36.4 | 60.8 | 69.8 | 43.2 | 47.6 | 43.6 | 54.3 | 51.9 |
| WiniQ | **16.9** | 59.1 | 37.0 | 60.8 | 69.7 | 43.2 | 48.9 | 44.9 | 56.2 | **52.5** |
| **W2A8** | | | | | | | | | | |
| RTN | 1.5e6 | 24.5 | 23.1 | 62.4 | 52.3 | 33.6 | 25.4 | 17.6 | 50.3 | 36.1 |
| GPTQ | 3.8e4 | 26.9 | 21.7 | 37.6 | 51.8 | 33.5 | 25.5 | 14.8 | 49.7 | 32.7 |
| SpinQuant | 3.8e2 | 31.0 | 20.1 | 45.5 | 54.6 | 33.8 | 26.7 | 16.2 | 50.9 | 34.9 |
| ParetoQ | 16.9 | 59.5 | 37.1 | 61.9 | 69.5 | 42.5 | 48.1 | 44.5 | 54.5 | 52.2 |
| WiniQ | **16.3** | 60.98 | 37.66 | 61.39 | 70.42 | 42.99 | 48.83 | 47.27 | 54.69 | **53.0** |
| LLaMA-3-3B | Wiki2 (↓) | ARC-e | ARC-c | BoolQ | PIQA | SIQA | HellaSwag | OBQA | WinoGrande | Avg. Accuracy (↑) |
| FP Model | 7.7 | 72.6 | 50.7 | 74.6 | 78.2 | 48.5 | 74.3 | 53.7 | 69.2 | 65.2 |
| **W1A8** | | | | | | | | | | |
| RTN | 7.3e7 | 25.0 | 22.5 | 37.6 | 49.5 | 32.9 | 25.0 | 27.1 | 49.6 | 33.7 |
| GPTQ | 5.9e7 | 26.6 | 21.9 | 37.6 | 52.5 | 33.7 | 25.6 | 15.0 | 49.4 | 32.8 |
| SpinQuant | 4.5e7 | 27.0 | 22.0 | 37.6 | 52.0 | 33.4 | 25.5 | 14.5 | 50.2 | 32.8 |
| ParetoQ | 15.7 | 63.1 | 39.1 | 63.0 | 70.9 | 42.6 | 50.6 | 46.7 | 56.5 | 54.1 |
| WiniQ | **14.8** | 64.2 | 40.5 | 63.6 | 71.1 | 42.8 | 52.7 | 49.4 | 57.3 | **55.2** |
| **W1.58A8** | | | | | | | | | | |
| RTN | 7.9e5 | 24.7 | 23.4 | 37.6 | 52.9 | 33.8 | 25.5 | 17.4 | 50.3 | 33.2 |
| GPTQ | 2.7e5 | 25.3 | 22.9 | 37.6 | 53.7 | 34.4 | 25.3 | 15.6 | 49.5 | 33.1 |
| SpinQuant | 3.1e3 | 26.6 | 23.4 | 39.1 | 52.7 | 34.7 | 25.3 | 16.4 | 48.5 | 33.3 |
| ParetoQ | 13.1 | 67.9 | 42.0 | 53.9 | 72.0 | 43.9 | 58.2 | 49.6 | 60.2 | 56.0 |
| WiniQ | **12.2** | 68.0 | 43.0 | 62.8 | 73.0 | 44.9 | 59.8 | 50.2 | 61.6 | **57.9** |

Table 13: We report the complete comparison results of **Qwen-3-1.7B** and **Qwen-3-0.6B** with 1–2 bit weights and 8-bit activations. We evaluate the perplexity on WikiText-2 and average zero-shot accuracy across eight QA tasks.

| Qwen-3-1.7B | Wiki2 (↓) | ARC-e | ARC-c | BoolQ | PIQA | SIQA | HellaSwag | OBQA | WinoGrande | Avg. Accuracy (↑) |
|---|---|---|---|---|---|---|---|---|---|---|
| FP Model | 16.2 | 68.9 | 41.0 | 78.9 | 71.7 | 45.1 | 59.6 | 38.5 | 61.6 | 58.2 |
| **W1A8** | | | | | | | | | | |
| ParetoQ | 46.5 | 42.0 | 26.2 | 61.4 | 59.8 | 40.0 | 29.1 | 28.9 | 50.8 | 42.3 |
| WiniQ | **45.9** | 42.4 | 26.4 | 61.4 | 59.6 | 40.3 | 29.0 | 30.1 | 49.7 | **42.4** |
| **W2A8** | | | | | | | | | | |
| ParetoQ | 22.2 | 52.9 | 32.6 | 62.4 | 65.1 | 42.7 | 41.2 | 32.8 | 53.1 | 47.8 |
| WiniQ | **21.8** | 53.1 | 33.9 | 62.7 | 64.7 | 42.3 | 41.0 | 35.0 | 53.1 | **48.2** |
| Qwen-3-0.6B | Wiki2 (↓) | ARC-e | ARC-c | BoolQ | PIQA | SIQA | HellaSwag | OBQA | WinoGrande | Avg. Accuracy (↑) |
| FP Model | 53.6 | 35.3 | 65.6 | 67.4 | 42.6 | 46.5 | 36.1 | 56.6 | 50.5 | 20.1 |
| **W1A8** | | | | | | | | | | |
| ParetoQ | 64.0 | 37.3 | 24.2 | 61.5 | 56.6 | 39.5 | 27.3 | 33.0 | 50.6 | 41.2 |
| WiniQ | **61.9** | 38.4 | 24.7 | 62.1 | 56.7 | 39.8 | 27.6 | 31.8 | 50.2 | **41.4** |
| **W2A8** | | | | | | | | | | |
| ParetoQ | **32.0** | 44.9 | 28.5 | 54.5 | 60.4 | 40.7 | 32.7 | 33.2 | 51.6 | 43.3 |
| WiniQ | 32.1 | 44.7 | 28.3 | 60.2 | 60.0 | 40.7 | 32.5 | 33.4 | 51.1 | **43.9** |

Table 14: We report the complete comparison results of applying our approach with the Hadamard Transform, on top of the QuEST baseline. We evaluate **LLaMA-3-1B** with 1–2 bit weights and 4-bit activations. We evaluate the perplexity on WikiText-2 and average zero-shot accuracy across eight QA tasks. HT refers to the Hadamard Transform.

| LLaMA-3-1B | Wiki2 (↓) | ARC-e | ARC-c | BoolQ | PIQA | SIQA | HellaSwag | OBQA | WinoGrande | Avg. Accuracy (↑) |
|---|---|---|---|---|---|---|---|---|---|---|
| FP Model | 9.6 | 64.8 | 42.5 | 64.8 | 74.8 | 44.8 | 64.4 | 50.2 | 61.5 | 58.5 |
| **W1A4** | | | | | | | | | | |
| QuEST | 42.9 | 41.0 | 26.9 | 61.4 | 59.2 | 40.0 | 30.1 | 30.7 | 50.3 | 42.4 |
| WɪɴɪQ w/ HT | **42.3** | 42.1 | 28.9 | 61.9 | 58.7 | 39.6 | 30.4 | 31.8 | 50.2 | **43.0** |
| **W2A4** | | | | | | | | | | |
| QuEST | 17.4 | 52.5 | 32.9 | 59.2 | 66.4 | 43.5 | 46.1 | 36.5 | 52.0 | 48.6 |
| WɪɴɪQ w/ HT | **16.9** | 52.8 | 33.1 | 62.2 | 65.5 | 42.0 | 46.3 | 38.9 | 53.4 | **49.3** |

Table 15: We report preliminary results on training a single model across multiple bit-widths (4, 3, 2, and 1 bit). Using **LLaMA-3-1B** as the base model, we follow the quantization setup of ParetoQ. For comparison, we adopt Matryoshka Quantization (Nair et al., 2025), a recent method for multi-bit training. In these experiments, activations are fixed to 16 bits. We evaluate performance using perplexity on WikiText-2 and average zero-shot accuracy across eight QA tasks.

| LLaMA-3-1B | Wiki2 (↓) | ARC-e | ARC-c | BoolQ | PIQA | SIQA | HellaSwag | OBQA | WinoGrande | Avg. Accuracy (↑) |
|---|---|---|---|---|---|---|---|---|---|---|
| Single bit-width training (ParetoQ), with total **640K** training steps | | | | | | | | | | |
| 1-bit | 16.9 | 57.3 | 36.2 | 62.4 | 69.1 | 41.9 | 48.3 | 45.1 | 55.0 | 51.9 |
| 2-bit | 12.5 | 64.8 | 41.7 | 62.8 | 73.1 | 44.0 | 56.6 | 52.0 | 58.5 | 56.7 |
| 3-bit | 10.9 | 65.3 | 41.9 | 64.2 | 73.8 | 43.9 | 61.3 | 47.7 | 59.5 | 57.2 |
| 4-bit | 10.3 | 67.4 | 43.4 | 64.4 | 74.8 | 44.4 | 63.5 | 50.4 | 61.4 | 58.7 |
| Multi-level bit-width training (Matryoshka Quantization), with total **240K** training steps | | | | | | | | | | |
| 1-bit | 44.0 | 51.5 | 34.4 | 56.1 | 64.3 | 39.9 | 38.8 | 35.9 | 51.3 | 46.5 |
| 2-bit | 20.2 | 55.6 | 35.6 | 62.4 | 66.0 | 41.4 | 44.0 | 44.9 | 54.1 | 50.5 |
| 3-bit | 13.8 | 63.0 | 39.6 | 59.8 | 71.1 | 43.6 | 54.5 | 48.4 | 56.6 | 54.6 |
| 4-bit | 12.6 | 63.3 | 40.8 | 63.3 | 72.1 | 44.2 | 56.6 | 50.0 | 59.0 | 56.2 |
| Multi-level bit-width training (Ours), with total **240K** training steps | | | | | | | | | | |
| 1-bit | 20.7 | 54.5 | 34.7 | 46.7 | 65.4 | 41.1 | 41.9 | 39.3 | 55.2 | 47.4 |
| 2-bit | 13.3 | 63.1 | 39.8 | 61.8 | 71.3 | 44.3 | 55.4 | 50.6 | 57.1 | 55.4 |
| 3-bit | 11.9 | 66.2 | 42.4 | 63.6 | 72.5 | 44.3 | 58.9 | 51.6 | 57.7 | 57.1 |
| 4-bit | 11.6 | 66.6 | 43.5 | 63.6 | 72.8 | 44.3 | 59.4 | 51.0 | 58.2 | 57.4 |

Table 16: We summarize the hyperparameters used for each setting. A batch size of $8 \times 8$ indicates 8 GPUs with a per-GPU batch size of 8. HT refers to the Hadamard Transform.

| Model | Weight | Activation | Learning rate | Batch size | Re-intialization interval $K$ | Interpolation scalar $\alpha$ | Standard deviation $\sigma$ |
|---|---|---|---|---|---|---|---|
| LLaMA-3-1B | 4-bit | 16-bit | $1 \times 10^{-5}$ | $8 \times 8$ | 40K | 0.2 | 0.0002 |
| LLaMA-3-1B | 3-bit | 16-bit | $1 \times 10^{-5}$ | $8 \times 8$ | 40K | 0.2 | 0.0002 |
| LLaMA-3-1B | 2-bit | 16-bit | $2 \times 10^{-5}$ | $8 \times 8$ | 80K | 0.2 | 0.001 |
| LLaMA-3-1B | 1.58-bit | 16-bit | $2 \times 10^{-5}$ | $8 \times 8$ | 80K | 0.2 | 0.001 |
| LLaMA-3-1B | 1-bit | 16-bit | $2 \times 10^{-5}$ | $8 \times 8$ | 60K | 0.4 | 0.001 |
| LLaMA-3-1B | 2-bit | 8-bit | $4 \times 10^{-5}$ | $8 \times 8$ | 60K | 0.2 | 0.001 |
| LLaMA-3-1B | 1.58-bit | 8-bit | $4 \times 10^{-5}$ | $8 \times 8$ | 80K | 0.2 | 0.001 |
| LLaMA-3-1B | 1-bit | 8-bit | $4 \times 10^{-5}$ | $8 \times 8$ | 60K | 0.4 | 0.001 |
| LLaMA-3-3B | 1.58-bit | 8-bit | $4 \times 10^{-5}$ | $8 \times 8$ | 60K | 0.2 | 0.001 |
| LLaMA-3-3B | 1-bit | 8-bit | $4 \times 10^{-5}$ | $8 \times 8$ | 60K | 0.2 | 0.001 |
| Qwen-3-1.7B | 2-bit | 8-bit | $2 \times 10^{-5}$ | $8 \times 8$ | 80K | 0.1 | 0.0002 |
| Qwen-3-1.7B | 1-bit | 8-bit | $2 \times 10^{-5}$ | $8 \times 8$ | 80K | 0.1 | 0.0002 |
| Qwen-3-0.6B | 2-bit | 8-bit | $2 \times 10^{-5}$ | $8 \times 8$ | 80K | 0.1 | 0.0002 |
| Qwen-3-0.6B | 1-bit | 8-bit | $2 \times 10^{-5}$ | $8 \times 8$ | 80K | 0.1 | 0.0002 |
| LLaMA-3-1B w/ HT | 2-bit | 4-bit | $4 \times 10^{-5}$ | $8 \times 8$ | 80K | 0.2 | 0.0002 |
| LLaMA-3-1B w/ HT | 1-bit | 4-bit | $4 \times 10^{-5}$ | $8 \times 8$ | 80K | 0.1 | 0.0002 |
| LLaMA-3-1B w/ HT | 2-bit | 4-bit | $4 \times 10^{-5}$ | $8 \times 8$ | 80K | 0.2 | 0.0002 |
| LLaMA-3-1B w/ HT | 1-bit | 4-bit | $4 \times 10^{-5}$ | $8 \times 8$ | 80K | 0.1 | 0.0002 |

