# OpenReview forum: "WiniQ: Accelerating Quantization-Aware Training of LLMs around Saddle Points"
_ICLR.cc/2026/Conference — Submitted to ICLR 2026_

### Official Review · Reviewer_H6wj · 2025-10-22

**Soundness:** 3
**Presentation:** 4
**Contribution:** 4
**Rating:** 8
**Confidence:** 4

**Summary:**

This paper aims to address the slow convergence problem for QAT in LLMs, especially in low-bit settings. The authors first analyse the reason lies in the weights converging to a flat surface near saddle points. Then they propose a novel weight re-initialization technique and noise injection. The experimenal results show that the proposed method can speed-up the training process and improve the QAT quantized LLMs performance.

This research is interesting and meaningful for LLM quantization community. I lean to accept this paper.

**Strengths:**

Originality:
This paper presents a fresh perspective on the challenge of slow convergence in quantization-aware training (QAT). By identifying the flat surface phenomenon, it provides novel insights into the training dynamics of QAT for large language models (LLMs). The proposed WINIQ method effectively accelerates training and enhances model performance.

Quality:
The paper is well-written and clearly presented. Its motivation, problem identification, and proposed solution are well-grounded, reasonable, and meaningful.

Clarity:
The presentation is coherent and logically organized. Both the motivation and the empirical analysis are clear and persuasive.

Significance:
The contributions of WINIQ are of broad significance for both academic research and practical LLM deployment. It substantially improves the efficiency of QAT for LLMs, while the insights into saddle-point behavior open promising avenues for future research.

**Weaknesses:**

Main comments:
The authors only evaluate their method on 1B–3B language models. While I understand that quantization-aware training is computationally expensive, I am still curious whether this approach can scale to larger models, such as 13B–70B parameters.

Minor comments:
1. Line 18: “We find the key reason is that” → should be “We find that the key reason is that.”
2. Line 281: if i + 1(mod K) is zero → should be if (i + 1) mod K == 0 then.

**Questions:**

1.  Could the authors discuss the potential scalability challenges (e.g., memory usage, training stability, or quantization error propagation) and whether any architectural or algorithmic modifications would be needed to extend their method to such scales?

2. Line 18: The phrase “We find the key reason is that” should be revised to “We find that the key reason is that.”

3. Line 281: The expression “if i + 1(mod K) is zero” should be corrected to “if (i + 1) mod K == 0 then.”

---

> ### Author Response · Authors · 2025-11-21
> **Response to Reviewer H6wj**
>
> We thank the reviewer for carefully reading our paper and recognizing the novelty, clarity, and significance of our paper. Next, we respond to each comment from the reviewer.
>
> **Comment 1**: “The authors only evaluate their method on 1B–3B language models. While I understand that quantization-aware training is computationally expensive, I am still curious whether this approach can scale to larger models, such as 13B–70B parameters. ”
>
> **Response**: Thank you for the suggestion! Our method is not tied to any particular model size and can be applied directly to larger architectures. To explore scalability, we conducted a preliminary experiment on Llama-8B, training it with 2-bit weight quantization and 16-bit activations for 150K steps. Our approach achieves around a relative **1**% improvement over the state-of-the-art QAT baseline (ParetoQ, NeurIPS 2025).
> Below, we report zero-shot accuracy across eight QA tasks; We will include these results in the revised paper.
> |             | Avg. Accuracy | ARC-e | ARC-c | BoolQ | PIQA | SIQA | HellaSwag | OBQA | WinoGrande |
> | ----------- | :-----------: | :---: | :---: | :---: | :--: | :--: | :-------: | :--: | :--------: |
> | ParetoQ     |     64.7      | 75.9  | 50.2  | 75.0  | 78.5 | 48.5 |   72.3    | 49.6 |    68.0    |
> | WiniQ (Ours) |   **65.3**    | 76.2  | 52.0  | 76.9  | 78.4 | 48.5 |   72.7    | 50.4 |    67.7    |
>
> While we did not evaluate larger models due to the computation constraints and short period of the rebuttal, these results suggest that our approach can plausibly scale to larger sizes as well.
>
> **Question 1**: “Could the authors discuss the potential scalability challenges (e.g., memory usage, training stability, or quantization error propagation) and whether any architectural or algorithmic modifications would be needed to extend their method to such scales?”
>
> **Response**: Our approach introduces only negligible additional runtime and memory overhead beyond the base quantization method. Therefore, no architectural or algorithmic modifications are required when scaling to larger models. Our approach applies directly as long as the underlying model itself fits within the available training memory.
>
> We also note that as model size increases, the performance gap between full-precision and quantized models tends to shrink—a trend reported in prior work such as ParetoQ (NeurIPS 2025), because larger models have more redundancy and are inherently easier to quantize.
>
> **Response to other comments**: Thanks for the suggestions! We have fixed the typos in the paper.
>
> **References**
>
> Zechun Liu, Changsheng Zhao, Hanxian Huang, Sijia Chen, Jing Zhang, Jiawei Zhao, Scott Roy, Lisa Jin, Yunyang Xiong, Yangyang Shi, Lin Xiao, Yuandong Tian, Bilge Soran, Raghuraman Krishnamoorthi, Tijmen Blankevoort, Vikas Chandra. Paretoq: Scaling laws in extremely low-bit llm quantization. NeurIPS 2025.

---

> > ### Comment · Reviewer_H6wj · 2025-11-21
> > **Official Comment by Reviewer H6wj**
> >
> > Thank you for the authors’ responses and additional experiments. My main concerns have been fairly addressed. Therefore, I would like to maintain my score and recommendation.

---

> > > ### Author Response · Authors · 2025-11-21
> > > **Reply**
> > >
> > > We are pleased to have addressed the reviewer’s concerns and sincerely appreciate the valuable comments. Thank you for your responses and the recommendation.

---

### Official Review · Reviewer_wBNS · 2025-10-29

**Soundness:** 4
**Presentation:** 4
**Contribution:** 2
**Rating:** 6
**Confidence:** 5

**Summary:**

This paper introduces WiniQ, a method designed to accelerate Quantization-Aware Training (QAT). The article provides a detailed analysis of the slow convergence observed in traditional QAT, identifying the core reason: model weights tend to converge towards flat surfaces near saddle points, where a large fraction of the Hessian eigenvalues are concentrated around zero. The paper argues that under the Straight-Through Estimator (STE) approximation, the loss surface is flat along the direction from the quantized weights back to the original (float) weights. Consequently, the Hessian matrix of the loss during training possesses very small eigenvalues, leading to slow convergence. This issue becomes more severe as the quantization bit-width decreases. To address this phenomenon, the authors propose using regular weight interpolation to increase the eigenvalues of the Hessian matrix, complemented by injecting stochastic noise during the optimization process to accelerate training. The authors conduct extensive experiments to demonstrate the effectiveness of the Wini method in accelerating training across various QAT scenarios (primarily weight-only quantization), and also provide comparisons against a range of Post-Training Quantization (PTQ) methods and state-of-the-art (SOTA) QAT approaches.

**Strengths:**

1.  The theoretical analysis is comprehensive and novel. The authors' approach primarily focuses on the significant impact of the mathematical properties (magnitude, sign, eigenvalues, etc.) of the Hessian matrix of the loss function during neural network training. They provide a focused analysis on how quantization affects the Hessian matrix, addressing a profound and fundamental theoretical issue in neural networks.
2.  The logic is clear and well-structured. The paper starts with the premise of accelerating QAT, clearly presents the scenario, analyzes the problem, proposes the solution, and finally demonstrates the experimental results, making the argument easy to follow and convincing.
3.  The WiniQ method, despite its simplicity, proves highly effective for accelerating QAT, especially showing significant results in the 1-bit weight-only quantization scenario. It can also be readily combined with many existing works focusing on Hadamard transforms. The paper provides sufficient experimental validation. Although experiments are not conducted on very large models or with a vast amount of data, the paper validates the effectiveness of the Wini method across different quantization settings on a smaller scale and performs a series of ablation studies to confirm the effectiveness of its components.

**Weaknesses:**

1.  **Lack of Activation Quantization Analysis:** In the QAT domain, there is a general consensus that activations are significantly harder to quantize than weights, and achieving low-bit quantization for both weights and activations is crucial for practical hardware acceleration. While this paper provides comprehensive theoretical analysis, it focuses solely on the behavior of weights and lacks in-depth analysis of activation quantization. The existing experiments primarily use 16-bit and 8-bit activations, which are far from the low-bit regime explored for weights. **Although it is regrettable that the authors did not further explore activation quantization, this does not affect the internal consistency and completeness of the presented work.**
2.  **Insufficient Experimental Scale:** Although the high GPU resource requirements for training are acknowledged, Figure 5 shows a series of training curves where the compared ParetoQ curve still appears to have significant room for further decrease. While the proposed WiniQ curve shows a large initial drop in loss after interpolation, its subsequent decline rate is very low. This raises questions: As the training scale increases, will WiniQ exhibit further potential for loss reduction, or is its benefit primarily limited to the initial drop achieved through weight interpolation?
3.  **Universality Concerns Regarding Weight Quantization Bit-width:** Based on the paper's theoretical claims, the WiniQ method is expected to show greater benefits for lower weight quantization bit-widths. However, the paper does not extensively discuss the method's performance when the weight bit-width is higher. Figure 5 also shows that the performance gap between standard QAT and WiniQ diminishes as the weight bit-width increases. The authors should provide results comparing their method with others at higher weight bit-widths. If the gap is minimal for W4/W8 scenarios, this should be explicitly stated.
4.  **Issues with Writing, Presentation, and Minor Errors:** For example, the abstract and introduction should not simultaneously state that WiniQ can accelerate training by up to 4x *and* improve final performance by up to 8.8%, as acceleration assumes consistent performance, while performance improvement assumes equivalent training effort. Fundamentally, both relate to accelerating model convergence. Regarding presentation, the full-precision bar in Figure 3 and the ablation study in Section 4.3 are recommended to be moved from the appendix into the main text (as they support the paper's soundness), with other parts of the main text potentially shortened accordingly. The data in the "FP Model" rows in Table 1 are clearly incorrect and require correction.

**Questions:**

1.  **Definition of the Zero Eigenvalue Threshold:** The paper states, "Numerically, we regard the estimated eigenvalues in a range between -10⁻³ and 10⁻³ as zero eigenvalues." However, I feel this approach might be less robust than defining zero eigenvalues based on a proportion (e.g., relative to the maximum eigenvalue). Could the authors provide further justification or theoretical backing for the chosen threshold of 10⁻³? Specifically, can they demonstrate or argue that singular values smaller than 10⁻³ indeed correspond to directions where the gradient's contribution to weight updates is so negligible that it can be practically ignored?

2.  **Missing W1A16 Curve in Figure 5:** The W1A16 configuration is present in Table 1. Why is its training curve not included in Figure 5? Omitting it makes Figure 5 seem incomplete, especially since W1 (1-bit weight quantization) appears to be a key strength of this work highlighted by the authors.

3.  **Comparison with PTQ in Table 1:** Fundamentally, QAT is expected to outperform PTQ methods. Including PTQ results in the main experimental table (Table 1) might lead to some confusion or could be perceived as a way to emphasize the proposed method's performance gains. Would it be clearer to present QAT methods (including Wini) separately, perhaps only including PTQ comparisons in a dedicated section or a supplementary table?

---

> ### Author Response · Authors · 2025-11-21
> **Response to Reviewer wBNS (1/3)**
>
> We thank the reviewer for carefully examining our paper and providing constructive suggestions. We are glad that the reviewer thinks that our analysis is novel, our presentation is clear, and our results are significant. Below, we respond to each comment of the reviewer.
>
> **Comment 1**: “While this paper provides a comprehensive theoretical analysis, it focuses solely on the behavior of weights and lacks an in-depth analysis of activation quantization. The existing experiments primarily use 16-bit and 8-bit activations, which are far from the low-bit regime explored for weights.”
>
> **Response**: Thanks for the comment! We would like to clarify that our experiments also include 4-bit activation quantization, as reported in Table 4. In these settings, we apply our approach on top of the SoTA activation quantization method, QuEST (ICML 2025), which uses the Hadamard transform. Our approach improves performance by up to **2.8**% in the 4-bit activation regime, demonstrating that our method applies to low-precision activation quantization as well.
>
> We also observe that low-precision activation quantization leads to even slower convergence and earlier stagnation of model performance than higher-precision activations, as shown in Figure 5.
>
> To better understand this behavior, we further analyzed the Hessian spectrum for 8-bit and 4-bit activation quantization using Llama-1B with 1-bit weight quantization. We find that lower-precision activations (8-bit and 4-bit) result in (**23**% and **32**%) smaller-magnitude Hessian eigenvalues compared to 16-bit activations, respectively. This indicates a flatter loss landscape and explains the slower training under low-precision activation quantization. We will include these results in the revised paper.
>
> **Comment 2**: “Figure 5 shows a series of training curves where the compared ParetoQ curve still appears to have significant room for further decrease. While the proposed WiniQ curve shows a large initial drop in loss after interpolation, its subsequent decline rate is very low. This raises questions: As the training scale increases, will WiniQ exhibit further potential for loss reduction, or is its benefit primarily limited to the initial drop achieved through weight interpolation?”
>
> **Response**: For each experiment, we follow the setup in the baselines (ParetoQ, NeurIPS 2025) and use 20B tokens for training (240K training steps). Our approach is trained under the same training budget.
>
> We further doubled the training budget to 40B tokens (480K steps) for Llama-1B with 1, 2, and 3-bit weights and 16-bit activations. Up to 40B tokens, our approach maintains the advantage over ParetoQ, outperforming ParetoQ by **7**% in PPL on average. We report the results for further training in a table below and will include the results in the updated paper.
>
> | PPL | W1A16  |        | W2A16 |        | W3A16 |        |
> |:------------------|:------:|:------:|:-----:|:------:|:-----:|:------:|
> | Training budget (# tokens) | 20B    | 40B    | 20B   | 40B    | 20B   | 40B    |
> | ParetoQ (NeurIPS 2025) | 16.9   | 16.2 | 12.5  | 12.3   | 14.0  | 13.9 |
> | WiniQ (Ours)              | **15.3** | **14.7** | **11.9** | **11.8** | **12.9** | **12.8** |

---

> ### Author Response · Authors · 2025-11-21
> **Response to Reviewer wBNS (2/3)**
>
> **Comment 3**: “The paper does not extensively discuss the method's performance when the weight bit-width is higher. Figure 5 also shows that the performance gap between standard QAT and WiniQ diminishes as the weight bit-width increases. The authors should provide results comparing their method with others at higher weight bit-widths. If the gap is minimal for W4/W8 scenarios, this should be explicitly stated.”
>
> **Response**: We clarify that the improvement of our approach is more significant in precision below 3-bit. At higher bit-widths, such as 4-bit, existing quantized models already perform very close to the full-precision model, leaving limited room for improvement. As shown in Table 1, our approach does not degrade performance at higher precision, and we will emphasize this point more explicitly in the paper.
>
> To quantify this, we compute the performance gap between quantized and full-precision models for both our approach and the state-of-the-art baseline (ParetoQ, NeurIPS 2025) across multiple bit-width settings. On average, our approach reduces this gap by **17**%. For 4-bit quantization, the gap itself is already small; therefore, the absolute improvement is also small.
> |   Performance gap to full-precision   | Llama-1B, W1A16 |          | Llama-1B, W1.58A16 |          | Llama-1B, W2A16 |          | Llama-1B, W3A16 |          | Llama-1B, W4A16   |          |
> | ----------------------------------------- | ----------------- | ------ | -------------------- | ------ | ----------------- | -------- | ------------------- | -------- | --------------------- | -------- |
> |                                             |         PPL         | Acc. |          PPL           | Acc. |         PPL         | Acc. | PPL                 | Acc. | PPL                   | Acc. |
> |                 SoTA method                 |         7.3         |   6.6    |          4.4           |   3.8    |         2.9         | 1.8      | 1.3                 | 1.3      | 0.7                   | 0.2      |
> |                    Ours                     |         5.7         |   5.9    |          3.3           |   2.9    |         2.3         | 1.8      | 1.3                 | 0.7      | 0.6                   | 0.1      |
> |             Relative reduction  of the gap            |       21.92%        |  10.61%  |         25.00%         |  23.68%  |       20.69%        | 0.00%    | 0.00%               | 46.15%   | 14.29%                | 50.00%   |
> | **Performance gap to full-precision** | **Llama-1B, W1A8**  |          | **Llama-1B, W1.58A8**  |          | **Llama-1B, W2A8** |          | **Llama-3B, W1A8**  |          | **Llama-3B, W1.58A8** |          |
> |                                             |         PPL         | Acc. |          PPL           | Acc. |         PPL         | Acc. | PPL                 | Acc.      |    PPL                 | Acc.       |
> |                 SoTA method                 |        13.7         |   10.3   |          8.6           |   6.6    |         7.3         | 6.3      | 8                   | 11.2     | 5.4                   | 9.3      |
> |                    Ours                     |        12.3         |   9.5    |          7.3           |    6     |         6.7         | 5.5      | 7.1                 | 10       | 4.5                   | 6.6      |
> |             Relative reduction  of the gap              |       10.22%        |  7.77%   |         15.12%         |  9.09%   |        8.22%        | 12.70%   | 11.25%              | 10.71%   | 16.67%                | 29.03%   |
>
> **Comment 4**: “The abstract and introduction should not simultaneously state that WiniQ can accelerate training by up to 4x and improve final performance by up to 8.8%, as acceleration assumes consistent performance, while performance improvement assumes equivalent training effort.”
>
> **Response**: We will separate the two sentences and state that the performance improvement is under the same training budget.
>
> **Comment 5**: “Regarding presentation, the full-precision bar in Figure 3 and the ablation study in Section 4.3 are recommended to be moved from the appendix into the main text.”
>
> **Response**:  We will move the evaluation of the full-precision model in Figure 8 to Figure 3. We will also move the full results of the ablation study to Section 4.3 of the paper.
>
> **Comment 6**: “The data in the 'FP Model' rows in Table 1 are clearly incorrect and require correction.”
>
> **Response**: Thanks for spotting the typo. We will correct the results of the full precision model in Table 1.

---

> ### Author Response · Authors · 2025-11-21
> **Response to Reviewer wBNS (3/3)**
>
> **Question 1**: “Definition of the Zero Eigenvalue Threshold”
>
> **Response**: We follow the common practice in numerical linear algebra and treat eigenvalues with magnitude below a small tolerance as effectively zero. In our setting, a threshold of $10^{-3}$ is already three orders of magnitude smaller than the largest scale of the Hessian eigenvalues. We also tested a relative threshold, $10^{-3}$ times the largest magnitude of Hessian eigenvalue, and obtained the same count of zero eigenvalues.
>
> **Question 2**: “Missing W1A16 Curve in Figure 5”
>
> **Response**: The W1A16 curve is already shown in Figure 1. Due to space constraints, we did not repeat it in Figure 5. If given an additional page in the final revision, we will include the W1A16 curve there as well.
>
>  **Question 3**: “Comparison with PTQ in Table 1”
>
> **Response**: We will separate the baseline results of PTQ in Table 1 to clearly demonstrate the categories of each baseline method.
>
> **References**
>
> Zechun Liu, Changsheng Zhao, Hanxian Huang, Sijia Chen, Jing Zhang, Jiawei Zhao, Scott Roy, Lisa Jin, Yunyang Xiong, Yangyang Shi, Lin Xiao, Yuandong Tian, Bilge Soran, Raghuraman Krishnamoorthi, Tijmen Blankevoort, Vikas Chandra. Paretoq: Scaling laws in extremely low-bit llm quantization. NeurIPS 2025.
>
> Andrei Panferov, Jiale Chen, Soroush Tabesh, Roberto L. Castro, Mahdi Nikdan, and Dan Alistarh. Quest: Stable training of llms with 1-bit weights and activations. ICML 2025.

---

### Official Review · Reviewer_tGJH · 2025-11-01

**Soundness:** 2
**Presentation:** 1
**Contribution:** 2
**Rating:** 2
**Confidence:** 5

**Summary:**

This paper proposes a gradient estimator for QAT LLMs.
It combines cyclic reinitialization and noise injection.
The idea is inspired by the observation of empirical loss Hessian geometry.

**Strengths:**

+ The idea is clearly presented.
+ Certain empirical observations are novel and interesting (see below).

**Weaknesses:**

- Although the empirical observation of loss Hessian gives some interesting results (Fig. 4), there is very little further analysis to study its generality or underlying mechanisms.
- Because of the above limitation, the proposed method, one that combines a zeroth-order noise injection with some pseudo-second-order perturbation by interpolation, still lack adequate theoretic or empirical support to establish either necessity or sufficiency.
- Presented ablation is not actually true ablation that establishes each component's necessity, but rather, a hyperparameter tuning of each component independently.  To establish necessity, you should study $\alpha=0$ and $\sigma=0$; to do proper hyperparameter tuning, you should search the joint $(\alpha, \sigma)$ space, not independently.

**Questions:**

* I find the non-monotonic behavior of Hessian spectral radius along the segment from $W$ to $Q(W)$ (Fig. 4) rather interesting, is this qualitatively general in all cases?  Is there any quantitative relationship about the location and magnitude of the maximal spectral radius as a function of model and quantization data type?
* I still do not have a clear picture of the empirical results that motivated the formulation of the technique.  Could you replot Fig. 4 with 2 joint independent variables, i.e. $(\alpha, \sigma)$?  Also, in addition to the indirect spectral radius of Hessian, could you also plot the expected gradient magnitude as the dependent variable?
* Due to the stochasticity from finite batch size, there should be an interaction between batch size and $\sigma$, what is such a relationship?
* Two concepts that should be separately studied are at times confused here: the symmetry (saddleness) and magnitude (flatness) of the Hessian eigenspectrum.  The authors characterized the loss landscape of QAT as both high in saddleness and flatness, is this both true?  Looking at Fig. 3, clearly the flatness is much greater in low precision case than in high precision, but I do not see any significant trend in the symmetry of the distribution.  Could you also include a column of full-precision training in Fig. 3?
* The cyclic reinitialization introduces the complexity of its schedule, how should the schedule be determined?  In particular, could you do an ablation study, in which you perform Fig. 4 analysis along the course of one training, and branch into different training instances with reinitialization at different times and compare across them on how fast they converge--if the intuition behind your method is correct, training instances that scheduled reinitialization at the most peaky Fig. 4 should win over those with less peaky ones.

---

> ### Author Response · Authors · 2025-11-21
> **Response to Reviewer tGJH (1/4)**
>
> We thank the reviewer for carefully examining our paper and recognizing that our presentation is clear and that our empirical observations are interesting. The reviewer requested additional analysis and ablation studies on the observations in Figure 4. We now respond to each comment of the reviewer below.
>
> **Comment 1**: “Although the empirical observation of loss Hessian gives some interesting results (Fig. 4), there is very little further analysis to study its generality or underlying mechanisms.”
>
> **Response**:  We are glad the reviewer found Figure 4 intriguing. We emphasize that this phenomenon is general. Across bit-widths from 1 to 4 bits, we also observe that the magnitude of the Hessian eigenvalues increases when evaluating interpolated weights between $W$ and $Q(W)$. This observation motivates our re-initialization technique by moving weights toward regions of higher curvature, which leads to faster training. More intuitively, this interpolation reduces the distance between $W$ and $Q(W)$, which aligns with the motivation of prior work such as QuEST (ICML 2025) to make quantized training more effective.
>
> We additionally measure the Hessian eigenvalue magnitudes ($\max_i \|\lambda_i\|$) for 1–4 bits when re-initializing to different interpolation positions (varying $\alpha$). We measure the Hessian with regard to model weights (including the learnable quantization scales). Using Llama-1B at 80K steps, we find that $\alpha = 0.4$ often produces the largest increase in Hessian eigenvalue magnitude. We will include these results in the updated paper.
> | $\max_i \|\lambda_i\|$ | $\alpha=0.0$ | $\alpha=0.2$ |     $\alpha=0.4$ | $\alpha=0.6$ | $\alpha=0.8$ | $\alpha=1.0$ |
> | ---------------------- | :------------: | :------------: | :----------------: | :------------: | :------------: | :------------: |
> | 1-bit                  | 0.93 $\pm$ 0.3 | 1.20 $\pm$ 0.1 | 1.72 $\pm$ 0.3 | **1.83** $\pm$ 0.1 | 1.24 $\pm$ 0.1 | 1.13 $\pm$ 0.1 |
> | 2-bit                  | 1.64 $\pm$ 0.5 | 2.45 $\pm$ 0.4 | **3.09** $\pm$ 0.4 | 2.65 $\pm$ 0.5 | 2.42 $\pm$ 0.5 | 2.05 $\pm$ 0.5 |
> | 3-bit                  | 6.24 $\pm$ 0.4 | 6.79 $\pm$ 0.4 | **7.70** $\pm$ 0.3 | 6.44 $\pm$ 0.3 | 6.68 $\pm$ 0.3 | 6.38 $\pm$ 0.2 |
> | 4-bit                  | 7.13 $\pm$ 0.4 | 7.31 $\pm$ 0.4 | **8.06** $\pm$ 0.5 | 7.61 $\pm$ 0.5 | 7.66 $\pm$ 0.5 | 7.12 $\pm$ 0.6 |
>
> We would also like to highlight that our paper already provides extensive Hessian-spectrum analysis across quantization levels: 1–3 bits in Figure 3, 4 bits in Figure 6, and full precision in Figure 8 of Appendix B. These results consistently show that weights (especially at low bit-widths) converge toward flat regions near saddle points, and that the loss surface becomes substantially flatter in lower precision. This offers a new insight into why low-precision quantized training exhibits slow convergence and early stagnation, which is an aspect not well-examined in prior work.

---

> ### Author Response · Authors · 2025-11-21
> **Response to Reviewer tGJH (2/4)**
>
> **Comment 2**: “Because of the above limitation, the proposed method, one that combines a zeroth-order noise injection with some pseudo-second-order perturbation by interpolation, still lacks adequate theoretic or empirical support to establish either necessity or sufficiency.”
>
> **Response**: As discussed above, our experiments show that interpolation consistently moves the weights into regions of higher curvature across different bit precisions. This indicates that the effect is driven by a general phenomenon rather than isolated cases, providing empirical support for the effectiveness of weight interpolation.
>
> Another technique, noise injection, computes gradients at weights perturbed by random noise. This is inspired by prior theoretical works (Jin et al., ICML 2017). Under a local second-order approximation around a first-order stationary point, this perturbation introduces a stochastic term proportional to the product between the Hessian and the random vector. Thus, this speeds up training around saddle points.
>
> We further evaluated the gradient norms (relative to the norm of weights) and Hessian eigenvalue magnitudes ($\max_i \|\lambda_i\|$) under different noise-injection levels $\sigma$ during training.
> - Using Llama-1B at 80K steps, we observe that noise injection tends to yield models with slightly larger gradient norms than standardly trained models.
> - Moreover, the model trained by noise injection tends to exhibit smaller negative eigenvalues (i.e., larger absolute values) compared to standard training.
>
> These empirical results also correlate with improved training speed of noise injection.
>
> |           | Training Loss       | Relative gradient norm | $\max_i \|\lambda_i\|$ |                |                    |                    |                |                |
> | --------------- | ------------------- | ---------------------- | ---------------------- | -------------- | ------------------ | ------------------ | -------------- | -------------- |
> |                   |                     |                        | $\alpha = 0.0$         | $\alpha = 0.2$ | $\alpha = 0.4$     | $\alpha = 0.6$     | $\alpha = 0.8$ | $\alpha = 1.0$ |
> |   $\sigma = 0$    | 3.18 $\pm$ 0.29 | 0.011 $\pm$ 0.002    | 1.64 $\pm$ 0.5         | 2.45 $\pm$ 0.4 | **3.09** $\pm$ 0.4 | 2.65 $\pm$ 0.5     | 2.42 $\pm$ 0.5 | 2.05 $\pm$ 0.5 |
> | $\sigma = 0.0005$ | 3.08 $\pm$ 0.28 | 0.012 $\pm$ 0.002    | 3.03 $\pm$ 0.2         | 3.53 $\pm$ 0.2 | **3.72** $\pm$ 0.1 | 3.50 $\pm$ 0.2     | 3.40 $\pm$ 0.2 | 3.18 $\pm$ 0.2 |
> | $\sigma = 0.001$ | 3.12 $\pm$ 0.29 | 0.013 $\pm$ 0.001    | 3.45 $\pm$ 0.3         | 3.91 $\pm$ 0.2 | 3.95 $\pm$ 0.2     | **3.96** $\pm$ 0.2 | 3.55 $\pm$ 0.2 | 3.32 $\pm$ 0.7 |
> | $\sigma = 0.002$ | 3.13 $\pm$ 0.28 | 0.013 $\pm$ 0.003    | 3.53 $\pm$ 0.2         | 3.94 $\pm$ 0.1 | 3.74 $\pm$ 0.1     | **3.85** $\pm$ 0.2 | 3.49 $\pm$ 0.2 | 3.53 $\pm$ 0.2 |
>
> We ground our contributions in empirical analysis. Establishing convergence guarantees in this setting is challenging because key assumptions used in prior theory may not hold. For instance, gradient Lipschitzness is hard to justify: straight-through estimators assign the identical gradients to all weights mapped to the same quantized value, which may cause abrupt gradient changes and violate smoothness assumptions. Therefore, we focus on empirical evidence. Developing formal guarantees for quantization-aware training remains an interesting direction for future research.
>
> **Comment 3**: “To establish necessity, you should study $\alpha=0$ and $\sigma=0$; to do proper hyperparameter tuning, you should search the joint space, not independently.”
>
> **Response**: Thanks for the suggestion!  We have included the $\sigma=0$ (no noise injection) in Table 4 in training Llama-1B in 1-bit. We further evaluated $\alpha=0$ (no weight re-initialization) in these experiments. Setting $\sigma=0$ and $\alpha=0$ results in worse performance—showing **4.5**% and **6.7**% higher PPL, respectively—compared to our full approach.
>
> Table 4 reports hyperparameter sweeps for each component. Conducting ablations on individual components is common practice in prior work (e.g., SpinQuant (ICLR 2025); QuEST (ICML 2025)). A full joint search is computationally prohibitive. Over the combined hyperparameter space, it would require $12\times3=36$ runs, each taking around 29 GPU hours.

---

> ### Author Response · Authors · 2025-11-21
> **Response to Reviewer tGJH (3/4)**
>
> **Question 1**: “I find the non-monotonic behavior of Hessian spectral radius along the segment from $W$ to $Q(W)$ (Fig. 4) rather interesting. Is this qualitatively general in all cases? Is there any quantitative relationship between the location and magnitude of the maximal spectral radius as a function of model and quantization data type?”
>
> **Response**: We additionally evaluated the magnitude of Hessian eigenvalues in the interpolation between $W$ to $Q(W)$ across bit widths from 1 to 4 bits. The results are consistent with those in Figure 4. **The results are reported in the response to Comment 1**. We will add these results to the updated paper.
>
> **Question 2**: “I still do not have a clear picture of the empirical results that motivated the formulation of the technique. Could you replot Fig. 4 with 2 joint independent variables, i.e., $(\alpha, \sigma)$? Also, in addition to the indirect spectral radius of Hessian, could you also plot the expected gradient magnitude as the dependent variable?”
>
> **Response**: We additionally evaluated the gradient norm and Hessian eigenvalues for various $\alpha$ and $\sigma$. **The results are described in the Response to Comment 2.**
>
> First, we consistently found that the Hessian eigenvalue magnitude typically peaks in the interpolation between $W$ and $Q(W)$, for both noise injection and no noise injection. Interpolation also reduces the distance between $W$ and $Q(W)$, which aligns with the motivation from prior work such as QuEST (ICML 2025).
>
> Second, we found that the models trained by noise injection tend to exhibit larger both the gradient norm and the Hessian eigenvalue magnitude than the standard trained one.
> We will add this result to the updated paper.
>
> **Question 3**: “Due to the stochasticity from finite batch size, there should be an interaction between batch size and $\sigma$, what is such a relationship?”
>
> **Response**: Thanks for the suggestion! We conducted an ablation study jointly varying the batch size and the noise standard deviation $\sigma$. Batch size was set to 64, 32, and 16, and $\sigma$ ranged from 0.0005 to 0.004. We evaluated these settings while training Llama-1B with 1-bit weight quantization and 16-bit activations for 240K steps.
>
> We observe that smaller batch sizes (e.g., 16) favor smaller optimal values of $\sigma$, while larger batches prefer larger $\sigma$ (typically around $0.001$). Batch size 64 produces the best overall performance; larger batches exceed our hardware memory limits. Accordingly, we use a batch size of 64 in the paper. We will include this discussion in the revised version.
>
> |  PPL             | $\sigma=0.0005$ | $\sigma=0.001$ | $\sigma=0.002$ | $\sigma=0.004$ |
> | :-----------: | :-------------: | :------------: | :------------: | :------------: |
> | Batch size 64 |      15.8       |    **15.3**    |      16.3      |      18.5      |
> | Batch size 32 |      16.2       |      16.1      |      17.0      |      19.1      |
> | Batch size 16 |      16.6       |      16.7      |      18.2      |      20.5      |

---

> ### Author Response · Authors · 2025-11-21
> **Response to Reviewer tGJH (4/4)**
>
> **Question 4**: “The authors characterized the loss landscape of QAT as both high in saddleness and flatness. Is this both true? Looking at Fig. 3, clearly the flatness is much greater in the low precision case than in the high precision, but I do not see any significant trend in the symmetry of the distribution. Could you also include a column of full-precision training in Fig. 3?”
>
> **Response**: The reviewer is correct that our results show models trained at lower precision converge to flatter regions of the loss landscape. We do not observe a substantial change in the “symmetry” of the eigenvalue distribution—that is, the proportions of positive and negative eigenvalues remain similar. However, this symmetry does not contradict our claims about slower convergence.
>
> The Hessian spectrum of full-precision is shown in Figure 8 of the Appendix. We will move them to Figure 3 in the updated paper.
>
> **Question 5**: “Could you do an ablation study, in which you perform Fig. 4 analysis along the course of one training, and branch into different training instances with reinitialization at different times and compare across them on how fast they converge--if the intuition behind your method is correct, training instances that scheduled reinitialization at the most peaky Fig. 4 should win over those with less peaky ones.”
>
> **Response**: We note that this ablation study is shown in Figure 9 of the Appendix. We branch the training into different instances and show the loss curves when using different $\alpha$ for interpolation. We found that $\alpha=0.4$ leads to the fastest training, which correlates with the highest magnitude of Hessian eigenvalue observed in Figure 4. We will also highlight this result in the main text.
>
> **References**
>
> Chi Jin, Rong Ge, Praneeth Netrapalli, Sham M. Kakade, and Michael I. Jordan. How to escape saddle points efficiently. ICML 2017.
>
> Zechun Liu, Changsheng Zhao, Hanxian Huang, Sijia Chen, Jing Zhang, Jiawei Zhao, Scott Roy, Lisa Jin, Yunyang Xiong, Yangyang Shi, Lin Xiao, Yuandong Tian, Bilge Soran, Raghuraman Krishnamoorthi, Tijmen Blankevoort, Vikas Chandra. Paretoq: Scaling laws in extremely low-bit llm quantization. NeurIPS 2025.
>
> Andrei Panferov, Jiale Chen, Soroush Tabesh, Roberto L. Castro, Mahdi Nikdan, and Dan Alistarh. Quest: Stable training of llms with 1-bit weights and activations. ICML 2025.
>
> Zechun Liu, Changsheng Zhao, Igor Fedorov, Bilge Soran, Dhruv Choudhary, Raghuraman Krishnamoorthi, Vikas Chandra, Yuandong Tian, and Tijmen Blankevoort. Spinquant: LLM quantization with learned rotations. ICLR 2025.

---

### Official Review · Reviewer_1LNG · 2025-11-10

**Soundness:** 2
**Presentation:** 3
**Contribution:** 2
**Rating:** 4
**Confidence:** 4

**Summary:**

This paper looks at why quantization-aware training (QAT) for large language models experiences significant performance degradation at very low bit-widths. Through Hessian spectrum checks, the authors show that weights get stuck on flat saddle regions, and they offer a simple fix—periodic linear interpolation plus light noise, that speeds things up. Experiments across 1–4-bit settings give modest gains in perplexity and QA accuracy. The key limitation is that only language-model tasks are tested, and the theoretical backing for the fix is thin.

**Strengths:**

- Novel Diagnostic Insight: the paper empirically link the slow convergence of sub-4-bit QAT to an increasingly flat Hessian spectrum dominated by near-zero eigenvalues and saddle regions.
- Simple yet Effective Algorithm: WiniQ adds only periodic linear interpolation plus light Gaussian noise; no second-order optimizer overhead and low compute cost.
- Clear Ablation: shows both interpolation and noise injection are essential; provides intuitive α and σ sensitivity curves.

**Weaknesses:**

- Limited Theoretical Justification: no formal convergence proof or saddle-escape guarantee for the proposed re-initialization schedule; empirical arguments dominate.
- The periodic linear-interpolation strategy functions more like a regularizer that keeps the optimizer out of local minima, but it lacks strong theoretical guarantees.
- The paper claims improved “training efficiency”; yet for QAT, the practical value of faster training is marginal—what truly matters to practitioners is inference speed and final accuracy.
- Limited Scope: results are restricted to Wiki-2 perplexity and QA tasks; no evidence on summarization, coding, or reasoning benchmarks.
- Incremental Gains: the improvements at some bit-widths reported in Table 1 are modest and incremental rather than breakthrough.

**Questions:**

- Could you clarify the exact computational overhead of WiniQ in wall-clock seconds rather than step ratios, including any extra memory traffic from re-initialization?
- Regarding the periodic re-initialization step: since the interpolation magnitude α is fixed and applied only every K iterations, have you analyzed whether the algorithm could still diverge if the loss surface curvature suddenly becomes sharp again later in training, and if so, what mechanism prevents such instability?

---

> ### Author Response · Authors · 2025-11-21
> **Response to Reviewer 1LNG (1/3)**
>
> We thank the reviewer for carefully reading the paper and recognizing the key insight and the effectiveness of our algorithm. We respond to each comment from the reviewer below.
>
> **Comment 1**: “Limited Theoretical Justification: No formal convergence proof or saddle-escape guarantee for the proposed re-initialization schedule. Empirical arguments dominate.”
>
> **Response**: In this work, we ground our contributions in empirical analysis and demonstrate performance gains in practical scenarios. Existing theoretical guarantees (e.g., Jin et al., ICML 2017) rely on assumptions that often do not hold under quantization, making it challenging to develop these results for our scenario. For example, gradient Lipschitzness is difficult to justify: straight-through estimators assign identical gradients to all weights mapped to the same quantized value, leading to abrupt gradient changes that violate smoothness assumptions. For these reasons, we focus on empirical evidence.
>
> Note that our study provides new insight from the optimization perspective, which connects quantized training to saddle-point problems. This is a perspective that has not been well-explored in prior work. Developing rigorous guarantees for quantization-aware training remains an interesting direction for future research.
>
> **Comment 2**: “The paper claims improved training efficiency. Yet, for QAT, what truly matters to practitioners is inference speed and final accuracy.”
>
> **Response**:  We agree with the reviewer and emphasize that our approach also achieves substantially higher final accuracy than state-of-the-art quantization methods, as shown in Figure 5 and Tables 1–3. Particularly, under the same training budget, our approach consistently outperforms SoTA quantized training algorithms by up to **8.8%** in quantization performance, across **16** settings of bit-widths and quantization methods. We will highlight these accuracy gains more clearly in the abstract and introduction.
>
> Regarding inference speed, our approach matches prior approaches because it uses the same quantization methods used in works including ParetoQ (NeurIPS 2025) and QuEST (ICML 2025). Inference speed depends on the chosen quantization method and bit-width, and our results demonstrate that our approach is compatible with a range of these methods, allowing practitioners to select configurations that meet their inference-speed requirements.
>
> **Comment 3**: “Results are restricted to Wiki-2 perplexity and QA tasks; No evidence on summarization, coding, or reasoning benchmarks.”
>
> **Response**: Our approach is generic and extends naturally to other evaluation settings. In a preliminary study, we conducted evaluations on reasoning tasks, comparing our method with a state-of-the-art baseline (ParetoQ, NeurIPS 2025).
>
> We trained Llama-1B-Instruct on the OpenMathReasoning dataset (Moshkov et al., 2025) with a maximum sequence length of 8K tokens, using 1-bit weight quantization and 16-bit activations for 15K steps. The table below reports test accuracy on GSM8K and MATH. The results show our approach yields around a 5% relative improvement over the baseline.
>
> As a remark, even strong quantized training baselines still exhibit relatively weak math reasoning performance. Existing quantization methods—such as ParetoQ (NeurIPS 2025), QuEST (ICML 2025), and SpinQuant (ICLR 2025)—primarily report perplexity and zero-shot QA results, and our evaluations follow this convention. Improving reasoning in low-precision models is not the main focus of our work, and it may require specially designed training strategies such as reinforcement learning.  Developing such training methods tailored to this setting is an interesting direction for future research.
>
> | Test Accuracy (%)  | GMS8K | MATH |
> | ------------------ | :---: | :--: |
> | ParetoQ            | 11.78 | 2.14 |
> | WiniQ (Ours) | 12.55 | 2.29 |

---

> ### Author Response · Authors · 2025-11-21
> **Response to Reviewer 1LNG (2/3)**
>
> **Comment 4**: “The improvements at some bit-widths reported in Table 1 are modest and incremental.”
>
> **Response**: We clarify that our approach can improve the quantization performance across all bit-width settings. The improvement is more significant in precision below 2-bit. This is because, for higher bit-widths, such as 4-bit and 3-bit, the existing quantized model performance is already close to the full-precision model. Thus, the room for further improvement is small.
>
> To further quantify the effect, we compare the performance gap between the quantized model and the full-precision model achieved by our method versus the state-of-the-art method (ParetoQ, NeurIPS 2025). Our approach reduces this gap by **17%** relative to ParetoQ, showing a meaningful improvement even when absolute gains appear smaller at higher bit-widths.
>
> |   Performance gap to full-precision   | Llama-1B, W1A16 |          | Llama-1B, W1.58A16 |          | Llama-1B, W2A16 |          | Llama-1B, W3A16 |          | Llama-1B, W4A16   |          |
> | ----------------------------------------- | ----------------- | ------ | -------------------- | ------ | ----------------- | -------- | ------------------- | -------- | --------------------- | -------- |
> |                                             |         PPL         | Acc. |          PPL           | Acc. |         PPL         | Acc. | PPL                 | Acc. | PPL                   | Acc. |
> |                 SoTA method                 |         7.3         |   6.6    |          4.4           |   3.8    |         2.9         | 1.8      | 1.3                 | 1.3      | 0.7                   | 0.2      |
> |                    Ours                     |         5.7         |   5.9    |          3.3           |   2.9    |         2.3         | 1.8      | 1.3                 | 0.7      | 0.6                   | 0.1      |
> |             Relative reduction  of the gap            |       21.92%        |  10.61%  |         25.00%         |  23.68%  |       20.69%        | 0.00%    | 0.00%               | 46.15%   | 14.29%                | 50.00%   |
> | **Performance gap to full-precision** | **Llama-1B, W1A8**  |          | **Llama-1B, W1.58A8**  |          | **Llama-1B, W2A8** |          | **Llama-3B, W1A8**  |          | **Llama-3B, W1.58A8** |          |
> |                                             |         PPL         | Acc. |          PPL           | Acc. |         PPL         | Acc. | PPL                 | Acc.      |    PPL                 | Acc.       |
> |                 SoTA method                 |        13.7         |   10.3   |          8.6           |   6.6    |         7.3         | 6.3      | 8                   | 11.2     | 5.4                   | 9.3      |
> |                    Ours                     |        12.3         |   9.5    |          7.3           |    6     |         6.7         | 5.5      | 7.1                 | 10       | 4.5                   | 6.6      |
> |             Relative reduction  of the gap              |       10.22%        |  7.77%   |         15.12%         |  9.09%   |        8.22%        | 12.70%   | 11.25%              | 10.71%   | 16.67%                | 29.03%   |

---

> ### Author Response · Authors · 2025-11-21
> **Response to Reviewer 1LNG (3/3)**
>
> **Question 1**: “Could you clarify the exact computational overhead of WiniQ in wall-clock seconds rather than step ratios, including any extra memory traffic from re-initialization?”
>
> **Response**: The additional runtime introduced by WiniQ is negligible relative to the base training cost. To further substantiate this, we measured wall-clock time in seconds on a machine with 26 CPUs and one A100 GPU.
> - For re-initialization, it only involves element-wise addition between two weight matrices. On Llama-1B, we perform re-initialization up to three times; This takes up to 0.3 seconds, compared to 28.7 hours for the full training run.
> - For noise injection, each iteration samples Gaussian noise and performs an element-wise addition with the weights. On Llama-1B, this costs 0.004 seconds per iteration—less than 1% of the time for a forward-backward pass (0.43 seconds).
>
> Our approach introduces no additional memory overhead. On Llama-1B, our approach uses the same peak GPU memory as the baseline (ParetoQ), which is 78.8 GB. This is because memory consumption is dominated by the forward and backward passes, where our techniques do not increase the memory. We will clarify this point in the revised paper.
>
> **Question 2**: “Regarding the periodic re-initialization step: since the interpolation magnitude $\alpha$ is fixed and applied only every $K$ iterations, have you analyzed whether the algorithm could still diverge if the loss surface curvature suddenly becomes sharp again later in training, and if so, what mechanism prevents such instability?”
>
> **Response**: Applying multiple re-initializations is designed to handle the cases when the training becomes slow again: After one re-initialization, our approach applies the re-initialization again after $K$ steps. This is not performed frequently: The interval $K$ varies from one-sixth to one-third of the total training steps. The precise hyperparameters are reported in Table 11. One may also design a schedule for $\alpha$, such as an exponentially decaying scheme analogous to a learning-rate schedule. We use a fixed $\alpha$ for simplicity and note that it already leads to significant improvement over the baselines.
>
> We also note that the loss curvature information accumulated by AdamW is preserved, since our re-initialization does not change the second-order states in the optimizer. This helps ensure training remains stable and prevents divergence following each re-initialization.
>
> **References**
>
> Chi Jin, Rong Ge, Praneeth Netrapalli, Sham M. Kakade, and Michael I. Jordan. How to escape saddle points efficiently. ICML 2017.
>
> Zechun Liu, Changsheng Zhao, Hanxian Huang, Sijia Chen, Jing Zhang, Jiawei Zhao, Scott Roy, Lisa Jin, Yunyang Xiong, Yangyang Shi, Lin Xiao, Yuandong Tian, Bilge Soran, Raghuraman Krishnamoorthi, Tijmen Blankevoort, Vikas Chandra. Paretoq: Scaling laws in extremely low-bit llm quantization. NeurIPS 2025.
>
> Panferov, Andrei, Jiale Chen, Soroush Tabesh, Roberto L. Castro, Mahdi Nikdan, and Dan Alistarh. Quest: Stable training of llms with 1-bit weights and activations. ICML 2025.
>
> Zechun Liu, Changsheng Zhao, Igor Fedorov, Bilge Soran, Dhruv Choudhary, Raghuraman Krishnamoorthi, Vikas Chandra, Yuandong Tian, and Tijmen Blankevoort. Spinquant: Llm quantization with learned rotations. ICLR 2025.
>
> Ivan Moshkov, Darragh Hanley, Ivan Sorokin, Shubham Toshniwal, Christof Henkel, Benedikt Schifferer, Wei Du, Igor Gitman. AIMO-2 Winning Solution: Building State-of-the-Art Mathematical Reasoning Models with OpenMathReasoning dataset. 2025

---

### Author Response · Authors · 2025-11-23
**Summary of responses and revisions (1/2)**

Thanks to the chairs for handling our submission. We understand the current situation and appreciate your additional efforts in assessing our work. Given the length of the reviews, we provide a summary of the current review situation to help the decision process.

We are glad to find that reviewers appreciate the novelty of our findings (1LNG, tGJH, wBNS, H6wj), the effectiveness of our approach (1LNG, wBNS, H6wj), the significance of the results (wBNS, H6wj), and the clarity of our presentation (1LNG, tGJH, wBNS, H6wj).

Below, we summarize the main comments from each reviewer and our responses (W indicates the Weaknesses, and Q indicates the Questions).

**Theoretical guarantees** (1LNG, W1&2; tGJH, W2): We emphasize that quantization-aware training remains a largely practical and empirically driven area. Prior work provides a limited understanding of why this training often converges slowly or exhibits early stagnation in performance. Our work contributes a new insight by connecting these problems to saddle points through empirical Hessian analyses. Developing theoretical guarantees is challenging because assumptions in existing theory may not hold under quantization. For instance, straight-through estimators can produce abrupt gradient changes that violate smoothness assumptions. Theory in this area is an interesting future direction; Our work provides an empirical foundation to help guide such efforts.

**Clarifying improvements in final accuracy** (1LNG, W3&5; wBNS, W3): Besides improved training efficiency, our approach consistently improves final accuracy over SoTA quantization methods across 16 settings, for weight in 1–4 bits and activation in 4–16 bits. The reviewers questioned that the gains at certain bit-widths appear modest. This is because, at higher precision like 4-bit, existing quantized models already perform very close to the full-precision model, leaving limited room for improvement. To better quantify the significance, we also measure the performance gap between the quantized and full-precision model. Our approach reduces the gap by 17% on average compared to SoTA baselines.

**Additional evaluation tasks** (1LNG, W4): We added evaluations on reasoning tasks, including GSM8K and MATH, and showed that our approach outperforms the SoTA baseline. We note that even the best quantization methods exhibit relatively weak math reasoning performance, and improving the performance on these tasks is an open question. Thus, we primarily evaluate on language tasks, following common practice in prior quantization work.

**Further analysis of Hessians** (tGJH, W1&Q1&Q2): The main concern of reviewer tGJH is the generality of the observation in Figure 4. We conducted requested evaluations (Q1&Q2) to show additional evidence for the underlying mechanism of our techniques. First, for weight interpolation, we consistently observe increased Hessian eigenvalue magnitudes in interpolation, across models trained at 1–4 bits and under different noise levels. This replicates the pattern in Figure 4 and empirically supports this technique. Second, for noise injection, we observe that noise-injection-trained models tend to exhibit smaller negative Hessian eigenvalues (larger absolute values) and larger gradient norms, supporting its effect of accelerated training. We also clarify that weight interpolation reduces the distance between quantized and full-precision weights, consistent with the motivation discussed in related work (e.g., QuEST, ICML 2025).

**Necessity of each component and other ablation studies** (tGJH, W2-3&Q3-5): We evaluated the effect of removing each component of our method and found that both contribute substantially to the final performance. We also conducted additional ablations requested by the reviewer, noting that most of them were already included in the submission. For Q3, we jointly varied the batch size and noise scale $\sigma$. We found that larger batch sizes tend to favor larger optimal values of $\sigma$. For Q4 and Q5, the requested Hessians and training curves were already included in the original submission. We have now moved the ablation studies into the main text and highlighted them in the revision.

---

> ### Author Response · Authors · 2025-12-02
> **Summary of responses and revisions (2/2)**
>
> **Response to other comments**:
> - *Testing the additional runtime and memory* (1LNG, Q1): We evaluated the overhead of both components of our method and found that each adds less than 1% runtime. Both technique introduces no additional memory.
> - *Analysis of activation quantization* (wBNS, W1): Our initial submission included activation quantization down to 4 bits and showed performance gains over the SoTA method (QuEST, ICML 2025) in low-precision activation. We further evaluate Hessian eigenvalues under 16-, 8-, and 4-bit activation. We find that training in lower activation precision leads to smaller Hessian eigenvalue magnitudes, explaining the slower training.
> - *Increasing the training budget* (wBNS, W2): We doubled the training budget and found that our approach continued to deliver improvements over the SoTA baseline.
> - *Experiments on larger models* (H6wj, W1): We applied our approach to Llama-8B and showed improvement over the SoTA baseline.
>
> We have provided a point-by-point response to all reviewer comments and updated the manuscript accordingly, with revisions highlighted in blue. We also included an anonymous link to our code for reproducing our work in the updated paper.
>
> Based on the above, we believe we have thoroughly addressed the reviewers’ concerns. For reviewer 1LNG, we clarified the significance of our improvements and justified our choice of evaluation tasks and focus on empirical studies. For reviewer tGJH, we completed all requested analyses and strengthened the evidence supporting the generality of our findings (with many ablations already present in the original submission). We therefore believe there is a strong likelihood that the reviewers will update their evaluation. In addition, reviewer H6wj explicitly expressed support for acceptance in both the review and discussion. We hope the chairs take these into account when making the final decision. Thank you again for your careful handling of the process.

---

### Comment · Area_Chair_dqJ4 · 2025-11-28

Dear authors and reviewers,

Please remain professional and refrain from being influenced by the event. If anyone violates the rules, please let me know, and I will flag and report it to the Program Chairs.

Your AC

---

### Meta-Review · Area_Chair_d1x4 · 2026-01-07

**Summary:**

The authors propose a novel approach to accelerate quantization-aware training (QAT) for large language models (LLMs), the reviewers generally expressed concerns regarding the method's effectiveness, theoretical foundation, and the comprehensiveness of experimental validation. The main issues include a lack of solid theoretical support, limited scope of experimental validation (mainly focused on language model tasks), and insufficient attention to performance across different model sizes and quantization bit-widths.

**Reviewer Concerns:**

#### Addressed by the Rebuttal:

- **Improvement in Accuracy and Efficiency**: The authors provided additional evaluations in their rebuttal, demonstrating that their method improved the performance gap between quantized models and full-precision models, addressing the reviewers' doubts about performance enhancements.

#### Outstanding Concerns:

- **Insufficient Theoretical Foundation**: Despite the authors' attempts to clarify their position, reviewers still expressed concerns about the lack of theoretical guarantees. The proposed technical approach appears inadequate without robust theoretical analysis to support it.
- **Narrow Experimental Scope**: Several reviewers mentioned that the experimental validation mainly focused on a limited set of tasks, lacking assessments on larger models and broader tasks, which raises questions about the generalizability of the results.
- **Scalability Issues**: Reviewers were concerned about the applicability and scalability of the method, especially regarding potential challenges it might face with different model sizes and quantization levels.

**Reviewer Scores:**

- **Reviewer 1LNG**: Initially rated 4 (marginally below acceptance threshold). Considering the clarity in the rebuttal, they might adjust their score to 6 (acceptable) but still hold reservations.
- **Reviewer tGJH**: Rated 2 (reject). This reviewer is likely to maintain this stance due to dissatisfaction with the theoretical shortcomings and empirical validation limitations.
- **Reviewer wBNS**: Rated 6 (marginally above acceptance threshold). This reviewer may be influenced by the rebuttal to slightly change their view, but due to unresolved major theoretical issues, their stance may lean toward rejection.
- **Reviewer H6wj**: Rated 8 (accept). Their confidence in accepting the paper seems strong, likely unchanged due to the addressed concerns.

---

### Decision · Program_Chairs · 2026-01-26

Reject